# PI3K-Yap activity drives cortical gyrification and hydrocephalus in mice

Achira Roy[1], Rory M Murphy[1], Mei Deng[2], James W MacDonald[3], Theo K Bammler[3], Kimberly A Aldinger[1,2], Ian A Glass[2], Kathleen J Millen[1,2]*

[1]Center for Integrative Brain Research, Seattle Children's Research Institute, Washington, United States; [2]Division of Genetic Medicine, Department of Pediatrics, University of Washington, Washington, United States; [3]Department of Environmental and Occupational Health Sciences, School of Public Health, University of Washington, Washington, United States

**Abstract** Mechanisms driving the initiation of brain folding are incompletely understood. We have previously characterized mouse models recapitulating human *PIK3CA*-related brain overgrowth, epilepsy, dysplastic gyrification and hydrocephalus (Roy et al., 2015). Using the same, highly regulatable brain-specific model, here we report PI3K-dependent mechanisms underlying gyrification of the normally smooth mouse cortex, and hydrocephalus. We demonstrate that a brief embryonic Pik3ca activation was sufficient to drive subtle changes in apical cell adhesion and subcellular Yap translocation, causing focal proliferation and subsequent initiation of the stereotypic 'gyrification sequence', seen in naturally gyrencephalic mammals. Treatment with verteporfin, a nuclear Yap inhibitor, restored apical surface integrity, normalized proliferation, attenuated gyrification and rescued the associated hydrocephalus, highlighting the interrelated role of regulated PI3K-Yap signaling in normal neural-ependymal development. Our data defines apical cell-adhesion as the earliest known substrate for cortical gyrification. In addition, our preclinical results support the testing of Yap-related small-molecule therapeutics for developmental hydrocephalus.

DOI: https://doi.org/10.7554/eLife.45961.001

*For correspondence:
kathleen.millen@seattlechildrens.org

**Competing interests:** The authors declare that no competing interests exist.

## Introduction

The mammalian brain has evolved through multiple transitions between gyrencephaly and lissencephaly (*Lewitus et al., 2014*). Cortical expansion and gyrification have been implicated in the evolution of human cognition; and dysplastic gyrification is associated with numerous neurodevelopmental disorders including hydrocephalus (*Jiménez et al., 2014*; *Guerra et al., 2015*; *Gregory et al., 2016*; *Parrini et al., 2016*; *Borrell, 2018*). Yet, despite the high significance, the mechanisms which create a smooth or folded brain remain poorly understood.

*In vivo* mouse and ferret studies together with in vitro human organotypic slice analyses have identified several genetic, cell biological and biomechanical factors that contribute to gyrification (*Borrell, 2018*; *Llinares-Benadero and Borrell, 2019*). Stereotypic, *bona fide* cortical folding includes formation of gyri (away from ventricle) and sulci (proximal to ventricle) of differential thickness (gyri thicker than the sulci), adjacent to a predominantly unfolded apical ventricular lining (*Borrell, 2018*). Naturally gyrencephalic species undergo a sequence of developmental events referred to here as the 'gyrification sequence'. It begins with apical progenitor proliferation followed by differential expansion of secondary progenitors, comprising of intermediate precursors (IPs) and basal radial glia cells (bRGs). The expansion of progenitors is associated with focal modes of neuronal differentiation, and migration that is mediated by variable ECM stiffness (*Borrell, 2018*; *Llinares-*

*Benadero and Borrell, 2019*). Although differential progenitor expansion is central to gyrification, little is known about the initiating steps of this highly regulated process (*Gregory et al., 2016*).

We previously generated mice with activating mutations of *Pik3ca*, the catalytic subunit of the phosphoinositide 3-kinase (PI3K) enzyme, to model human brain overgrowth syndromes including megalencephaly and epilepsy (*Roy et al., 2015*). Since dysplastic cortical folding and developmental hydrocephalus are within the spectrum of PI3K-related brain overgrowth syndromes (*Mirzaa et al., 2012*; *Keppler-Noreuil et al., 2014*; *Jansen et al., 2015*), we have now used our tet-inducible activating *Pik3ca*$^{H1047R}$ mutant mice (*Roy et al., 2015*) to study mechanisms underlying these additional phenotypes. Hydrocephalus, affecting approximately 1 in 1000 births, is among the most common neurodevelopmental disorders with often devastating outcome (*Guerra et al., 2015*; *Tully et al., 2016*). It is characterized by abnormal expansion of brain ventricles (ventriculomegaly) and progressive accumulation of cerebrospinal fluid (*Jiménez et al., 2014*; *Guerra et al., 2015*). New therapeutic approaches are urgently needed since current treatment requires invasive surgeries with associated significant complications (*Khan et al., 2015*). PIK3CA-related hydrocephalus is a subtype of developmental hydrocephalus, caused by disrupted brain development associated with genetic abnormalities (*Tully and Dobyns, 2014*; *Tully et al., 2016*). Infantile hydrocephalus can also result from environmental insult, including intra-ventricular hemorrhage associated with prematurity (*Adamsbaum et al., 1998*; *Jiménez et al., 2009*). Despite identification of several contributing factors, the underlying cellular and molecular mechanisms that cause hydrocephalus remain largely unknown.

Here we report that Pik3ca activation in embryonic cortical progenitors during a critical two-day period was sufficient to drive cortical gyrification in mice. PI3K activation disrupted apical junctions and caused ectopic subcellular translocation of Yap leading to neural proliferation and gyrification, as well as abnormal ependymal development and hydrocephalus. Both the gyrification and hydrocephalus phenotypes were attenuated in the mutant mice by treatment with verteporfin (*US Food and Drug Administration, 2000*; *Schmidt-Erfurth and Hasan, 2000*; *Liu-Chittenden et al., 2012*), a nuclear Yap inhibitor.

These results demonstrate that the PI3K/Hippo-Yap pathway is finely tuned to regulate cell adhesion and proliferation along the apical lining of the forebrain to maintain the lissencephalic mouse brain. Subtle alterations in this pathway during the mid-neurogenic phase have dramatic consequences for the cytoarchitecture of forebrain ventricular linings and the interrelated processes of neurogenesis, gyrification and ependymal development.

## Results

### *Pik3ca* activating mutations caused gyrification of the normally lissencephalic mouse cortex

We identified striking gyrification of the hippocampus and neocortex in the embryonically induced *GFAP-cre;Pik3ca*$^{H1047R}$ mutant mice (*Figure 1a–l*; *Roy et al., 2015*) This mutant also recapitulates human PI3K-related developmental hydrocephalus, without any evidence of stenosis along the antero-posterior extent of the brain. In this model, a transgene encoding an activating *H1047R* mutation in the human *PIK3CA* gene has dual spatio-temporal regulation, such that the presence of both *cre* protein and doxycycline is required to activate the mutation in cre-positive neuronal progenitors (*Figure 1—figure supplement 1a,b*) (*Roy et al., 2015*). The *GFAP-cre* driver used in this study gets activated in neural progenitors at around embryonic day (E)13 (*Zhuo et al., 2001*; *Roy et al., 2015*). Around this early activation time, this *cre* line demonstrates a strong high-medial-low-lateral expression gradient in the forebrain, as well as an apical-low-basal-high gradient within the lateral neocortex, as seen at E14.5 (*Figure 1—figure supplement 1c*). These differential gradients decrease gradually with the progression of developmental age (*Figure 1—figure supplement 1d*). Embryonically induced postnatal (P)3 *GFAP-cre;Pik3ca*$^{H1047R}$ mutants (doxycycline: E0.5>P3) demonstrated highly convoluted medial tissue, with indistinct hippocampal morphology (*Figure 1b, d,f*) (*Roy et al., 2015*). This hippocampal folding phenotype was 100% penetrant with stereotypic gyral pattern in all mutants (n > 50) studied. 3-D models of P3 control and mutant hippocampi

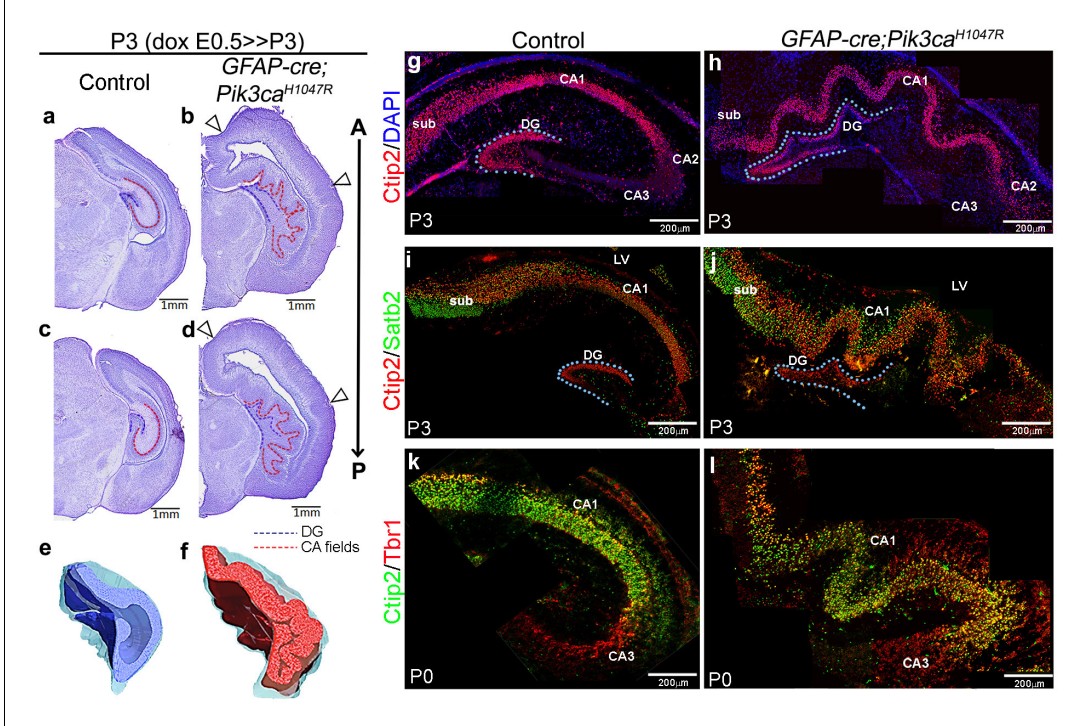

**Figure 1.** Embryonic induction of *Pik3ca*[H1047R] activating mutation causes cortical gyrification in mice. (a–d) Nissl-stained coronal sections of P3 control and *GFAP-cre;Pik3ca*[H1047R] mutant hemi-forebrains revealed hippocampal (dotted lines) and neocortical gyrification (open arrowheads), when induced embryonically. (e,f) 3D models of control and mutant hippocampi. (g–l) Gross patterning of all hippocampal substructures was intact, as shown by Ctip2, Satb2 and Tbr1 expression. Medial gyrification was restricted primarily to the mutant CA1 region. CA – cornus ammonis (red dotted line, a–d); sub – subiculum; DG – dentate gyrus (blue dotted line, a-d; g–j). Scale bars: 1 mm (a–d), 100 μm (n–s). See also *Figure 1—figure supplements 1–3*.
DOI: https://doi.org/10.7554/eLife.45961.002

The following figure supplements are available for figure 1:

**Figure supplement 1.** Genetic strategy for *Pik3ca*[H1047R] mouse model and expression of *GFAP-cre* line.
DOI: https://doi.org/10.7554/eLife.45961.003
**Figure supplement 2.** Characterization of true gyrification in *Pik3ca*[H1047R] mutant neocortex and hippocampus.
DOI: https://doi.org/10.7554/eLife.45961.004
**Figure supplement 3.** *Pik3ca*[H1047R] mutant demonstrates increase in ventricular length.
DOI: https://doi.org/10.7554/eLife.45961.005

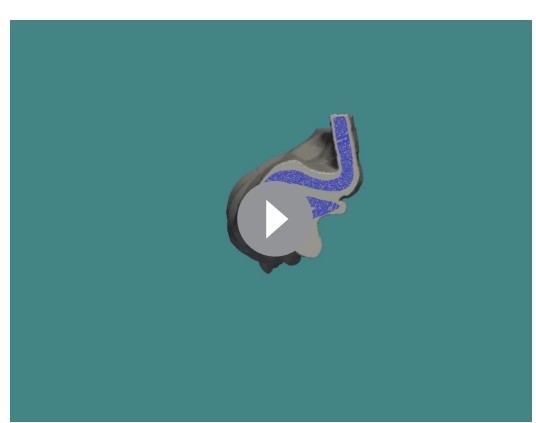

**Video 1.** 3D model of P3 control hippocampus.
DOI: https://doi.org/10.7554/eLife.45961.006

clearly demonstrate the differential anatomical features (*Video 1*, *Video 2*; still images in *Figure 1e,f*). Ventriculomegaly, indicating developing hydrocephalus, was also clearly evident in all of the mutant brains, seen from as early as E14.5 (*Figure 1b,d*; *Figure 1—figure supplement 2f,p–s*; *Figure 1—figure supplement 3a, c,e*). Less pronounced lateral neocortical folding was also evident with 100% penetrance, although variable in position (*Figure 1b,d*; *Figure 1—figure supplement 2f–i*). Both neocortical and hippocampal folds in the *Pik3ca*[H1047R] mutant mice followed the criteria of *bona fide* cortical gyrification (*Borrell, 2018*): a) folded pial surface and underlying layers, b) predominantly smooth apical surfaces adjacent to the lateral ventricle, c) differential thickness between gyri

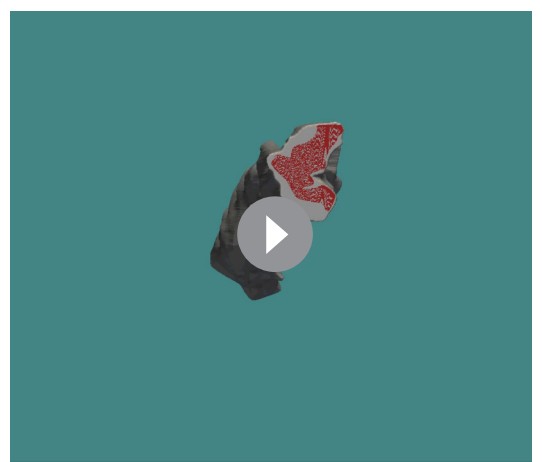

**Video 2.** 3D model of P3 *GFAP-cre;Pik3ca^{H1047R}* hippocampus.
DOI: https://doi.org/10.7554/eLife.45961.007

and sulci (*Figure 1—figure supplement 2f–i,p–s*). Notably, similar lateral and medial cortical gyrification was also documented in mice with *Pik3ca^{H1047R}* mutation activated by *Emx1-cre* (*D'Gama et al., 2017*). Together, these data demonstrate that Pik3ca activation is sufficient to cause cortical gyrification across the entire mouse dorsal telencephalon with the phenotype likely dependent on the time and regional gradients of *cre* activity, and regional differences in neurogenic periods.

## Focal increases in progenitors initiated *bona fide* cortical gyrification in *Pik3ca^{H1047R}* mutant

To determine if *Pik3ca^{H1047R}* mice modeled *bona fide* cortical gyrification seen in naturally gyrified mammals (*Borrell, 2018*), we assessed neural patterning and progenitor proliferation during embryogenesis. We focused on the developing hippocampus, where gyrification was most prominent in pattern and location. P3 *Pik3ca^{H1047R}* mutants showed normally patterned subdivisions of hippocampus proper (CA1-4) and dentate gyrus (DG; *Figure 1g–l*). Similar to the control littermates, Ctip2 expression was high in the mutant CA1 and DG, sparse in CA2 and absent in Tbr1$^+$ CA3. Ctip2 expression also revealed that the gyrification in mutants was primarily restricted to CA1, with possibly secondary folding evident in the DG. The *Pik3ca* activating mutation resulted in a significant increase in the lengths of P3 CA field (p<0.0001) and dentate gyrus (p=0.03), compared to the respective controls (*Figure 1—figure supplement 3a,b*). The first visual sign of reproducible cortical folding was observed at E16.5 in the *Pik3ca^{H1047R}* mutant dorso-lateral neocortex (*Figure 1—figure supplement 3e*).

Further histological analyses confirmed that *GFAP-cre;Pik3ca^{H1047R}* mutants replicated the coordinated neurogenic sequence seen in gyrencephalic mammals. Compared with control, both E14.5 and E16.5 mutants had significantly longer medial ventricular linings (p=0.0028 and p<0.0001 respectively; *Figure 1—figure supplement 3c–f*), suggesting early expansion of apical progenitors. Short BrdU pulse confirmed significantly higher proliferation (p<0.05) in mutants at E14,5, but not at E16.5, compared to respective controls (*Figure 2b–e*). Sox2$^+$ primary progenitors in the mutant CA1 ventricular-subventricular zone were significantly increased at E14.5 (p<0.05) and decreased at E16.5 (p<0.01). Tbr2$^+$ IPs) were unchanged at E14.5; but significantly increased at E16.5 (p<0.05), demonstrating a gradual increase in secondary progenitor pool *in lieu* of the primary progenitors (*Figure 2f–i*). This trend of higher progenitor number was maintained even postnatally, especially in the mutant gyral ventricular-subventricular zone (*Figure 2l–m'*). The number of Sox2$^+$Tbr2$^-$ bRGs in outer subventricular zone was not significantly different between control and mutant at E16.5 (*Figure 2i*). The mutant CA1 further demonstrated significantly higher neuronal differentiation at E16.5 (*Figure 2j,k*).

Birth-dating analysis showed normal neuronal fate specification in P0 mutant CA1 (*Figure 2—figure supplement 1a–c*). Although the mutant hippocampal subdivisions were grossly intact, there were minor migration defects as evidenced by a loosely packed *stratum pyramidale* (PL), reduced *stratum lacunosum-moleculare* (slm), and scattered Calbindin$^+$ pyramidal cell subtype in the gyral white matter (*Figure 2l',m'*; *Figure 2—figure supplement 1f–i,k,l*). Similar to our report of the neocortical phenotype (*Roy et al., 2015*), we observed a disrupted Nestin$^+$/GFAP$^+$ radial glial scaffold throughout mouse embryogenesis, starting as early as E14.5 (*Figure 2—figure supplement 1d–i*). A divergent fiber distribution was evident postnatally at each gyrus, similar to that seen in naturally gyrencephalic species (*Figure 2n–o'*) (*Fernández et al., 2016*; *Borrell, 2018*). Focally concentrated progenitor cells found in the P3 gyral ventricular-subventricular zone and the funneling effect of radial glial scaffold fibers most certainly amplified the earlier developmental disruptions leading to *bona fide* gyrification of the normally lissencephalic cortex.

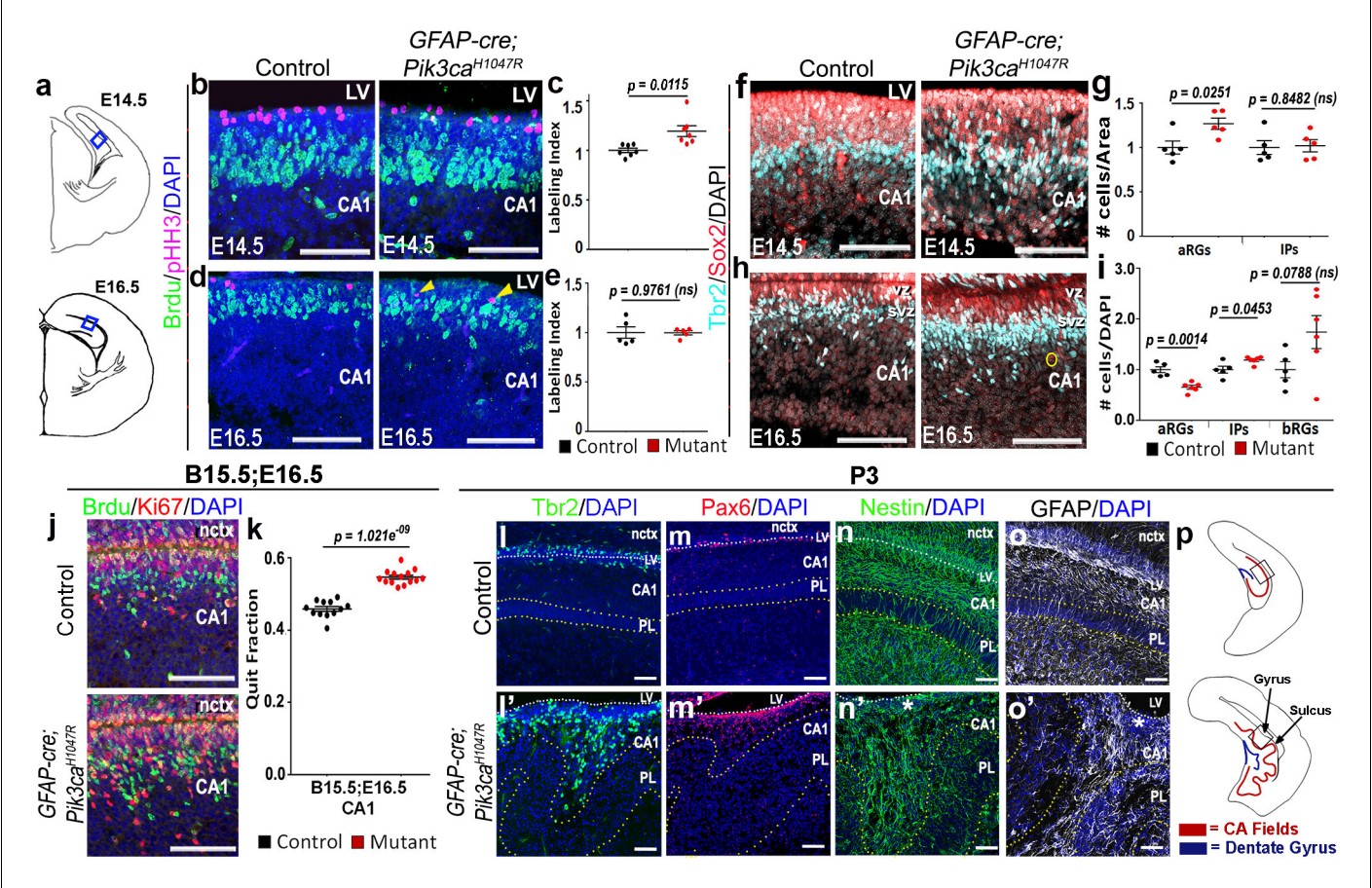

**Figure 2.** Altered neurogenesis in *Pik3ca^H1047R* mutant at embryonic and postnatal stages follows the stereotypic cortical 'gyrification sequence'. (**a**) Schematics of E14.5 and E16.5 coronal hemi-section, boxes represent region of interest (**b,d,f,h,j**). Images are oriented with lateral neocortex on the top, followed by lateral ventricle (LV) and then CA1 (**b,d,f,h,j,l–o'**). (**b–e**) Compared to control, E14.5 *Pik3ca^H1047R* mutants exhibited significantly higher labeling index, indicating higher proliferation rate. Though E16.5 mutant CA1 showed no overt change in labeling index, pHH3$^+$ cells marking the M-phase appeared to be ectopic (yellow arrowheads), indicating early mis-localization of dividing cells. (**f–i**) Compared with respective controls, E14.5 and E16.5 *GFAP-cre;Pik3ca^H1047R* mutants, induced from E0.5, showed significant increase in primary (apical Sox2$^+$Tbr2$^-$, aRGs) and secondary (Tbr2$^+$ IPs; basal Sox2$^+$Tbr2$^-$, bRGs) progenitor pools, in accordance with the gyrification sequence; yellow circle marks bRG. Data are normalized and represented as mean ± SEM in scatter plots; 2-tailed unpaired t-tests were performed (**c,e,g,i**). (**j,k**) Cell cycle exit during the E15.5-E16.5 period was significantly higher in the mutant CA1 than the control, as represented in mean ± SEM scatter plots (t = 10.16, degrees of freedom (df) = 21.73). (**l–m'**) Compared to controls, P3 mutant CA1 region showed focal increase in Pax6$^+$ and Tbr2$^+$ progenitors at the gyral ventricular-subventricular zone; the apical edge remained predominantly unfolded. (**n–o'**) Nestin$^+$ and GFAP$^+$ intermediate filaments in P3 were misoriented and divergent from focal points (asterisks) at the mutant gyri. (**p**) Schematics of P3 control and mutant hemi-sections; boxes depict regions shown in l-o'. Differences were considered significant at p<0.05; ns, not significant. nctx, neocortex; PL, pyramidal layer. Scale bars: 50 μm (**b,d,f,h,j**), 100 μm (**l–o'**). See also *Figure 2— figure supplement 1*.

DOI: https://doi.org/10.7554/eLife.45961.008

The following figure supplement is available for figure 2:

**Figure supplement 1.** *Pik3ca^H1047R* mutant demonstrates normal cell fate specification but disrupted neural scaffold and migration.

DOI: https://doi.org/10.7554/eLife.45961.009

## *Pik3ca* mutation-driven gyrification in mice had a short embryonic critical period

Although numerous mouse models of cortical gyrification exist, our highly regulatable model is an extremely valuable tool to dissect fundamental molecular mechanisms. Specifically, via temporally restricted doxycycline administration, we generated a time-series of *GFAP-cre;Pik3ca^H1047R* mutants to begin to dissect mechanisms underlying gyrification and other phenotypes (*Figure 3b–d,f–i*). Consistent with our previous study (*Roy et al., 2015*), postnatal induction of *Pik3ca^H1047R* allele

failed to cause overtly abnormal brain morphology (*Figure 3i*). By contrast, doxycycline induction from either E0.5 or E13.5 generated the most severely gyrified CA1 with identical stereotypic pattern (*Figure 3b–d,k*). This was not unexpected since despite doxycycline availability, the *cre* protein was produced in neural progenitors only from ~E13 (*Roy et al., 2015*). Pik3ca-related ventriculomegaly resulted broadly from mutation induction at any embryonic time point (*Figure 1b,d*; *Figure 1—figure supplement 2f,p*; *Figure 1—figure supplement 3a,c,e*; *Figure 3b–d,f–h*; *Figure 3—figure supplement 1c*).

Remarkably, a short 2 day window of doxycycline administration from E13.5 to (>) E15.5 was sufficient to completely recapitulate the highly stereotypic gyrification pattern seen in mutants with constitutive administration of doxycycline throughout development. Despite the identical gyrification pattern, the CA1 PL of P3 (doxycycline: E13.5 > E15.5) mutant was more compact than that of the constitutively activated mutants, suggesting a later role for activated Pik3ca in regulating the minor neuronal migration abnormalities detected in this model. Shorter periods of doxycycline treatment (E13.5 > E14.5, E14.5 > E15.5) within this critical period did not result in gyrification, despite mild cortical dysplasia and ventriculomegaly (*Figure 3—figure supplement 1b–e*). Induction beyond E15.5 (E15.5 > P3, E15.5 > E17.5, E17.5 > P0) caused a gradual attenuation of the gyrification phenotype and significantly smaller CA1 PL lengths in the respective P3 mutants (*Figure 3f–h,k*; *Figure 3—figure supplement 1a*). To further define gyrification mechanisms, we therefore focused our analysis of developmental events using minimally induced (doxycycline E13.5 > E15.5) mutant mice.

## Disordered junctional proteins at embryonic ventricular lining were the earliest presage of the mutant gyrification phenotype

The first obvious morphological sign of CA1 gyrification in minimally induced (doxycycline E13.5 > E15.5) mutants was a medial 'ripple' at E17.5 (*Figure 3—figure supplement 1f–n*). This was accompanied by dispersed radial glial fibers, loosely packed Ctip2[+] PL with ectopic Calbindin[+] cells, as well as reduced *slm* marked by cell adhesion molecule L1 (*Figure 3—figure supplement 1o–v*). Since these phenotypes suggested that neuronal migration abnormalities might underlie the gyrification phenotype, we investigated localization of ECM protein Reelin in the *Pik3ca[H1047R]* mutant. Reelin is well-known for its role in cell migration; and its expression in Cajal-Retzius cells is typically required to direct normal lamination of pyramidal neurons in the development of CA fields (*D'Arcangelo et al., 1995*). Further, ectopic Reelin has been reported to contribute to cortical dysplasia in another mouse model of PI3K pathway overactivation (*Baek et al., 2015*). However, we found no evidence of ectopic Reelin[+] cells in either E17.5 or P2 mutant CA1 regions (*Figure 2—figure supplement 1m,n*; *Figure 3—figure supplement 1r,v*).

Focal increases in proliferation observed at the mutant ventricular-subventricular zone suggested that disruptions at the apical (ventricular) linings were fundamental to the gyrification phenotype. In control E16.5 and P3 mice, the neocortical and hippocampal ventricular linings are tightly juxtaposed (*Figure 4b–e*, *Figure 4—figure supplement 1b–d*). The normal juxtaposition of apical membranes was disrupted in E16.5 mutants, leading to the formation of loose gaps/bubbles along the edge (*Figure 4f–i*), which eventually resulted in completely unzipped ventricular linings postnatally (*Figure 4—figure supplement 1e–j*). Junctional proteins delineating the ventricular surface, including β−Catenin, N-Cadherin, ZO-1 (*Kadowaki et al., 2007*), were focally discontinuous and often ectopic in P3 mutants, compared to their littermate controls. Milder, yet clear disruptions of cell polarity were observed at E16.5 and E14.5 along the mutant medial apical membrane (*Figure 4f–i,o–r*).

## Focal increases in nuclear yap (nYap) and neural progenitors at the mutant ventricular edge corresponded to gyrification zones

Forebrain ependymal development is normally initiated in mouse during mid-gestation (*Jiménez et al., 2014*). Our analysis of ependymal markers (Yap, Vimentin, Six3) indicated clear abnormalities in mutant mice (*Schnitzer et al., 1981*; *Lavado and Oliver, 2011*; *Park et al., 2016*). Especially striking were alterations in the localization of Yap, a molecule that sits at the nexus of multiple signaling pathways and that is normally expressed in blood vessels and at the ZO1[+] apical membrane of differentiating ependymal cells (*Figure 5b,c,o–p,s,s'*). At P3, numerous ectopic Yap[+] cells were distributed distal to the mutant CA1 ventricular edge, particularly predominant in the developing gyri (*Figure 5e,i,m*). At E16.5, we observed focal concentrations of Yap[+] cells in the

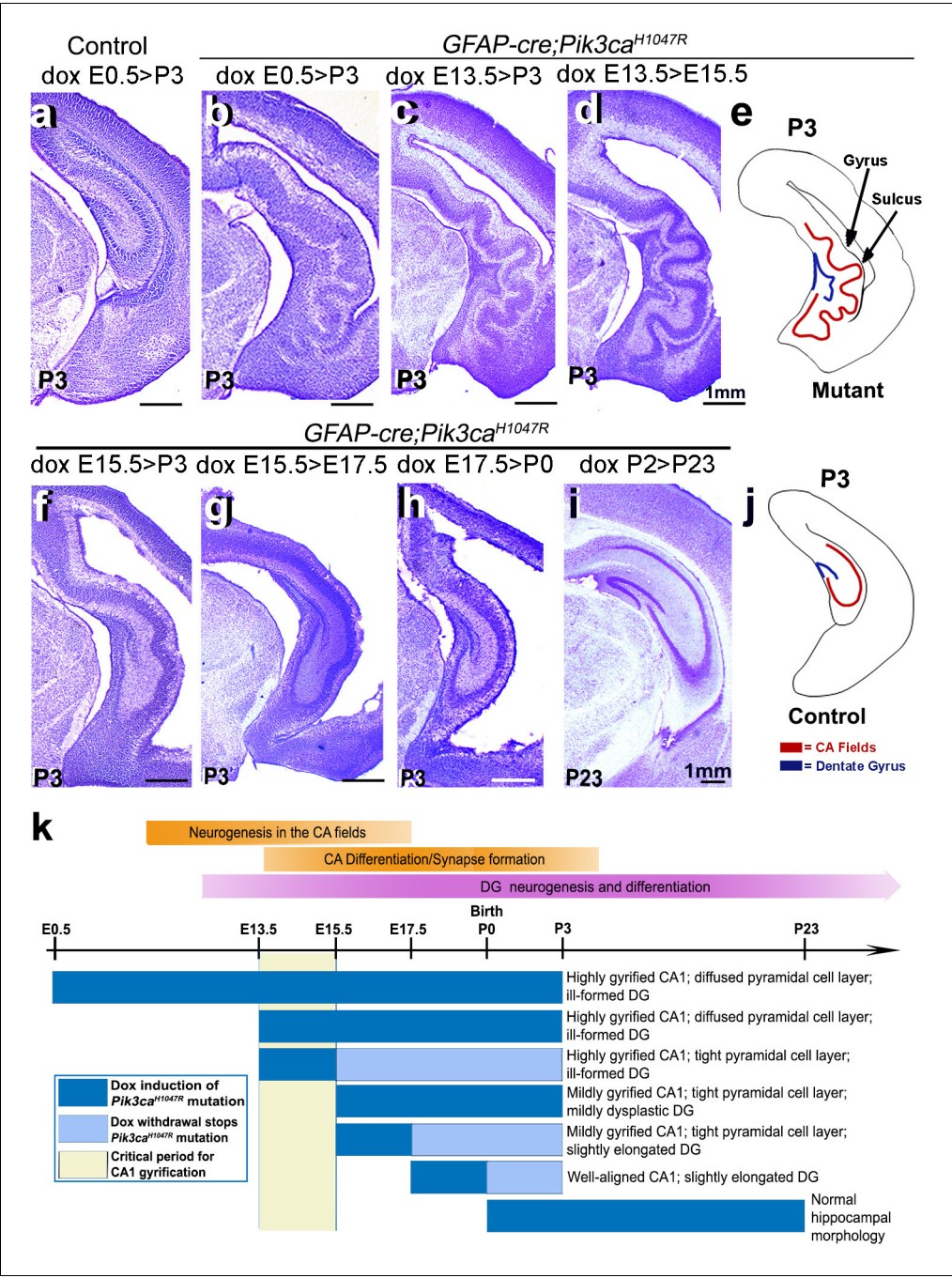

**Figure 3.** Non-random gyrification pattern in *Pik3ca^H1047R* mutant has a narrow embryonic critical period. (**a–j**) Nissl-stained coronal hemi-sections depicting hippocampal morphology, alongside schematics (**e,j**). Compared to control (**a**), *Pik3ca^H1047R* mutant hippocampus showed graded severity in gyrification, depending on the time and duration of mutation induction. The critical period of the most severe folding with a non-random folding pattern ends at E15.5 (**b–d**). Postnatal induction of *Pik3ca^H1047R* mutation is not effective to cause cortical gyrification (**i**). (**k**) Developmental timeline of mutation induction and the corresponding hippocampal gyrification phenotype. Scale bars: 1 mm (**a–d, f–i**). See also *Figure 3—figure supplement 1*.

DOI: https://doi.org/10.7554/eLife.45961.010

The following figure supplement is available for figure 3:

**Figure supplement 1.** Minimal induction of PI3K overactivation is sufficient to initiate cortical folding and altered neuronal migration in E17.5 CA1.

DOI: https://doi.org/10.7554/eLife.45961.011

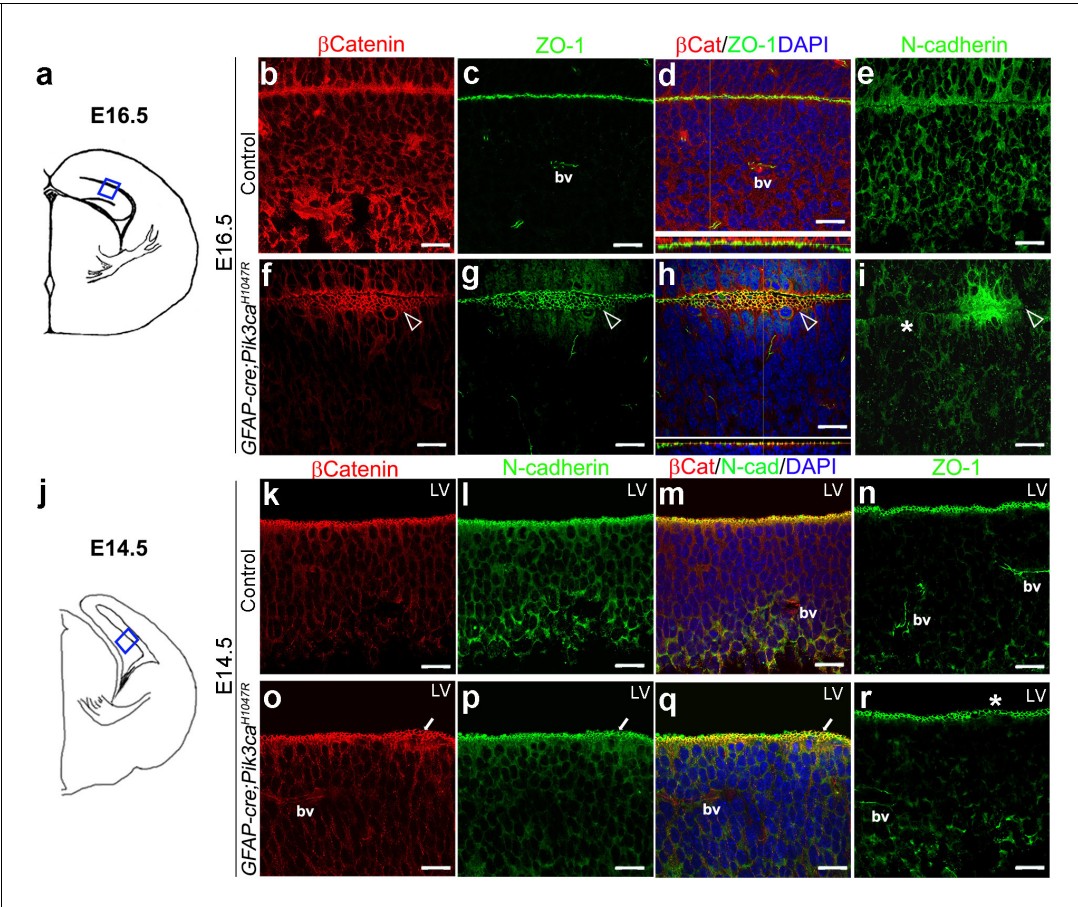

**Figure 4.** Embryonic induction of *Pik3ca*<sup>H1047R</sup> mutation causes early disruption of apical cell adhesion. (a,j) Schematic of E14.5 and E16.5 hemi-section showing area of interest. (b–i) Compared to control, E16.5 *Pik3ca*<sup>H1047R</sup> mutant, induced from E0.5, showed abnormal zippering of forebrain apical membranes (open arrowheads, (f–h), with XZ plane indicating subtle mis-localization of cell adhesion molecules β-Catenin and ZO-1 (d,h). N-Cadherin, that is normally expressed uniformly at the juxtaposition of the apical membranes (e), appeared to be clustered around the unzipped portion and sporadically absent (asterisk) in adjacent parts of the membrane junction (i). (k–r) Compared to control, CA1 ventricular lining of E14.5 mutant showed subtle disruption in the localization of β-Catenin, N-Cadherin, ZO1, as marked by white arrows (o–q) and asterisk (r). bv – blood vessels. Scalebars: 20 μm (b–i,k–r). See also *Figure 4—figure supplement 1*.

DOI: https://doi.org/10.7554/eLife.45961.012

The following figure supplement is available for figure 4:

**Figure supplement 1.** Disruption of apical cell adhesion caused by embryonic induction of PI3K overactivation persists postnatally.

DOI: https://doi.org/10.7554/eLife.45961.013

unzipped areas of the mutant ventricular lining; subtle focal disruption was evident as early as E14.5 (*Figure 5q,t*). Notably, Yap localization was predominantly cytoplasmic (cYap) and apical at all stages in control mice. In mutants however, we observed aberrant nuclear localization of Yap at all stages, especially evident at P3. The total number of nYap+ cells is significantly higher in P3 mutant sulcus and gyrus, compared with the controls (p<0.0001, *Figure 5l*). Magnified images of E16.5 and E14.5 mutant apical edges displayed a more frequent presence of nYap+ cells, compared to the respective controls (*Figure 5o',q',s',t'*). Disruption of ependymal development was also supported by altered expression of Vimentin and Six3 expression in the mutant (*Figure 5d,h,k*; *Figure 5—figure supplement 1a–d*).

## Inhibition of nYap attenuated gyrification and rescued ventriculomegaly

In neural stem cells, nYap drives proliferation and cYap acts to stabilize apical adherens junctions (*Park et al., 2016*; *Lavado et al., 2018*). We therefore hypothesized that focal nuclear mis-localization of Yap, from cytoplasm to nucleus, in the early mutant ventricular zone, drives focal over-

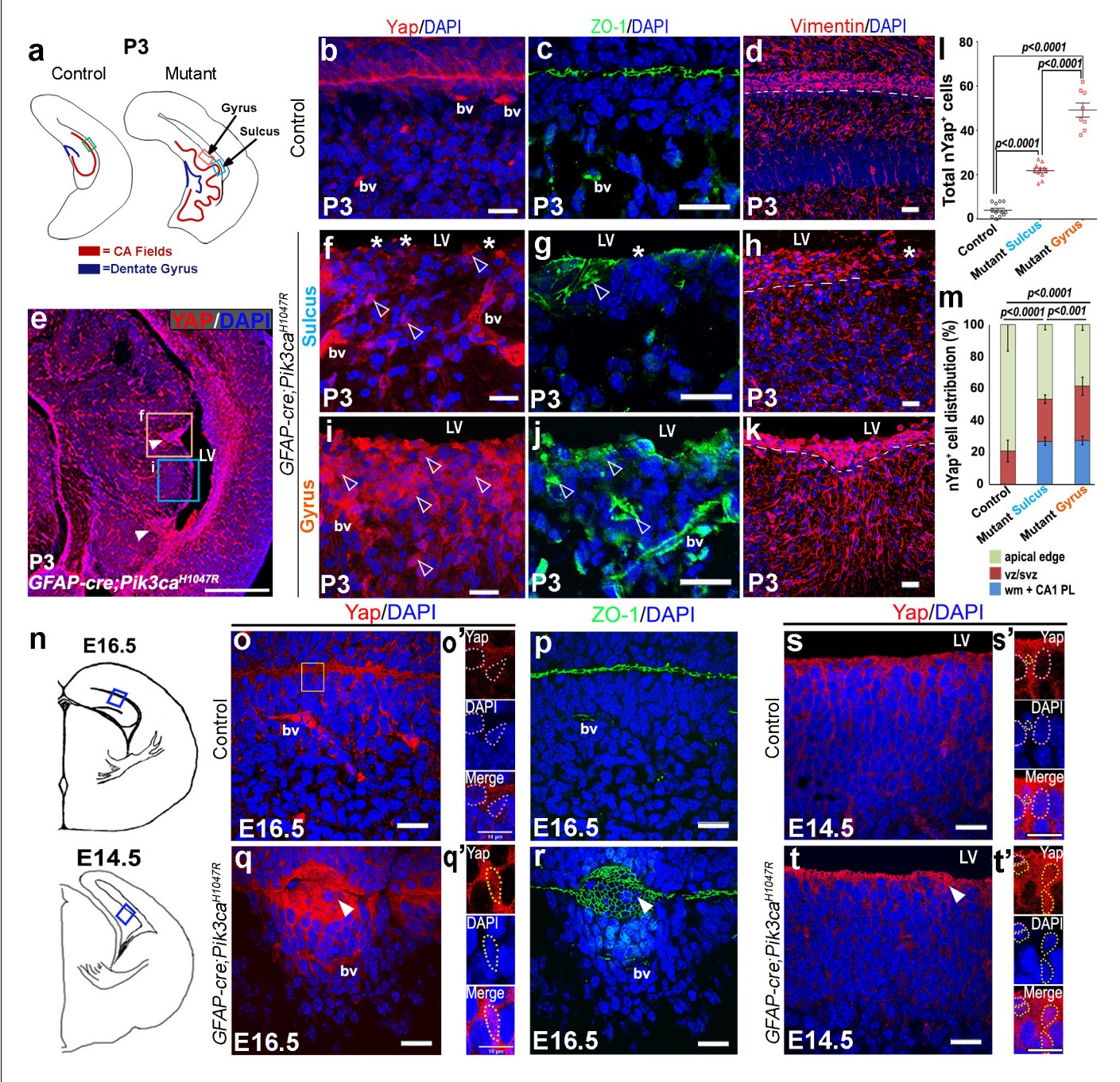

**Figure 5.** Embryonic induction of *Pik3ca*[H1047R] mutation disrupts early developing ependyma and induces increase in nYap+ cells. (a,n) Schematics of P3, E16.5 and E14.5 hemi-sections; boxed regions depict areas of interest. (b-d,o,o'p,s,s') Ependymal markers Yap and Vimentin, and junction protein ZO-1 are localized at the apical lining of P3, E16.5 and E14.5 control CA1 respectively. Additionally, Yap and ZO-1 mark the blood vessel membranes (bv), while Vimentin mark the hippocampal radial glia. (e) P0 mutant CA1 gyral ventricular lining showed presence of focal nYap-enriched zones (arrowheads, orange box), interspersed by Yap-sparse zones at the sulci (cyan box). (f,i) Increased Yap+ nuclei in P3 mutant gyri (open arrowheads) and Yap-sparse mutant sulcus edge (asterisks) indicated disrupted ependyma. (g,j) ZO-1 was ectopically expressed in P3 mutant gyral ventricular zone, with breaks in the sulcus areas. (h,k) Vimentin+ fibers appeared to be arranged in an hour-clock pattern at the gyri; sulci demonstrated more disoriented fibers and gaps (asterisk); dashed lines mark the basal ependymal edge. (l,m) Total number and distribution of nYap+ cells in the different CA1 subzones (binned as mono-layer apical edge, ventricular-subventricular zone (vz/svz), white matter (wm) and CA1 pyramidal layer (CA1 PL)) were significantly enhanced in P3 mutant, compared to control littermates (total counts: F = 179.1, df = 28; cell distribution: F = 137.8, df = 86). The control CA1 showed minimal existence of nYap+ cells beyond the vz/svz. Data are represented as mean ± SEM in scatter plots (l) or 100% stacked columns (m); one-way and two-way ANOVA were performed respectively. (o–q') Compared with control, E16.5 *Pik3ca*[H1047R] mutant showed abnormal zippering of

*Figure 5 continued on next page*

*Figure 5 continued*

forebrain apical membranes, combined with focal nYap expression (arrowheads). (**s–t'**) Compared with control, E14.5 mutant medial apical membrane showed subtle disruption of cell adhesion causing minor buckling/unevenness. Magnified images of E14.5 and E16.5 ventricular edge demonstrated higher number of nYap[+] cells in the mutant (yellow dotted lines) compared to controls (**o',q',s',t'**); some non-nYap[+] cells are marked with pink dotted lines for the purpose of comparison (**o',s',t'**). Scale bars: 20 μm (**b–d, f–k, o–t**), 500 μm (**e**), 10 μm (**o',q',s',t'**). See also *Figure 5—figure supplement 1*.
DOI: https://doi.org/10.7554/eLife.45961.014

The following figure supplement is available for figure 5:

**Figure supplement 1.** Six3[+] cells ectopically concentrate at the mutant gyral ventricular zone.
DOI: https://doi.org/10.7554/eLife.45961.015

proliferation and disrupts apical cell adhesion at nascent mutant gyri. To test this hypothesis, we administered nYap inhibitor verteporfin (*US Food and Drug Administration, 2000*; *Schmidt-Erfurth and Hasan, 2000*; *Liu-Chittenden et al., 2012*), by intraperitoneal injection to pregnant dams (doxycycline: E13.5 > E15.5; verteporfin: daily E13.5 > E18.5; *Figure 6a*). Morphological analysis at P0 confirmed that verteporfin treatment indeed attenuated the Pik3ca-related gyrification severity and completely rescued ventriculomegaly in 100% of the mutant brains, with little effect on control CA1 ventricular-subventricular zone morphology (*Figure 6b–g*, *Figure 6—figure supplement 1a–h,j*).

Using laser-capture method, we micro-dissected ventricular regions of the hippocampi from verteporfin treated and untreated mice, isolated RNA and performed RNA sequencing. Principal component analysis of the global transcriptional profiles of laser micro-dissected P0 CA1 ventricular-subventricular zone tissue (*Figure 6e–i*) confirmed that the untreated *Pik3ca* mutant samples (both gyrus and sulcus) were clearly distinct from the control samples. Verteporfin treatment had minimal effect on gene expression of control tissue but shifted the mutant gene expression profiles towards controls. Specifically, verteporfin treatment caused the mutant sulcus samples to cluster together with control samples, while treated mutant gyrus samples showed evidence of partial normalization. As expected, *Pik3ca* expression was higher in mutant samples, and suppressed post-verteporfin treatment, reflecting the complex feedback signaling in the PI3K pathway (*Carracedo and Pandolfi, 2008*). Gene set enrichment analysis confirmed that PI3K and Hippo-Yap signaling pathways were significantly differentially regulated in untreated mutants compared to controls, as expected (*Figure 6—figure supplement 2a–d*, *Figure 6—source datas 1* and *2*). Verteporfin treatment brought both PI3K and Hippo-Yap gene expression in mutants close to control levels, especially at the sulci. Notably, there was a considerable overlap in the gyral and sulcal gene expression profiles of our *Pik3ca*[H1047R] mutant mice and those mapped from ferret (*de Juan Romero et al., 2015*) – a naturally gyrencephalic mammal (*Figure 6—figure supplement 3a,b*). These suggest that our activating *Pik3ca* mutant gyrification model has significant physiological relevance to at least this naturally gyrencephalic species.

Immunohistochemical analyses of P0 verteporfin-treated mutants revealed normalized apical cell polarity in both the regions of attenuated sulci and gyri, with elimination of ectopic nYap[+] cells and rescue of contiguous apical ZO1 and cYap localization (*Figure 7b,c,h–i',n–o'*). Verteporfin normalized both nYap[+] and Six3[+] cell number and distribution in P0 mutant gyri and sulci (*Figure 6—figure supplement 1k,l*; *Figure 7h',n',t,u*). The 5 day acute verteporfin treatment of mutant mice resulted in streamlined Vimentin[+]/Nestin[+] radial glia, denser CA1 PL cell packing and reduction in the Tbr2[+] basal progenitor pool to almost control levels (*Figure 7d–g,j–m',p–s',v*). Restoration of P0 mutant ventricular cell polarity was readily apparent with β-Catenin and N-Cadherin expression, which also highlighted the close apposition of the neocortical and hippocampal ependymal surfaces (*Figure 7—figure supplement 1a–i'*). The effect of verteporfin on tissue architecture was identifiable even early, as studied in E16.5 *Pik3ca*[H1047R] mutants and control littermates (*Figure 7—figure supplement 2a–c*). Compared to the untreated mutants, E16.5 verteporfin-treated mutants (verteporfin: E13.5 > E16.5) demonstrated increased alignment of junctional proteins, normalized progenitor pool numbers and reduced occurrence of unzipped apical membranes (*Figure 7—figure supplement 2d–g'*). TUNEL assay showed no significant difference in apoptotic cell number between E16.5 control and mutant CA1, both in untreated and in verteporfin-treated conditions, indicating that cell death is not an important factor influencing the mutant phenotypes (*Figure 7—figure supplement*

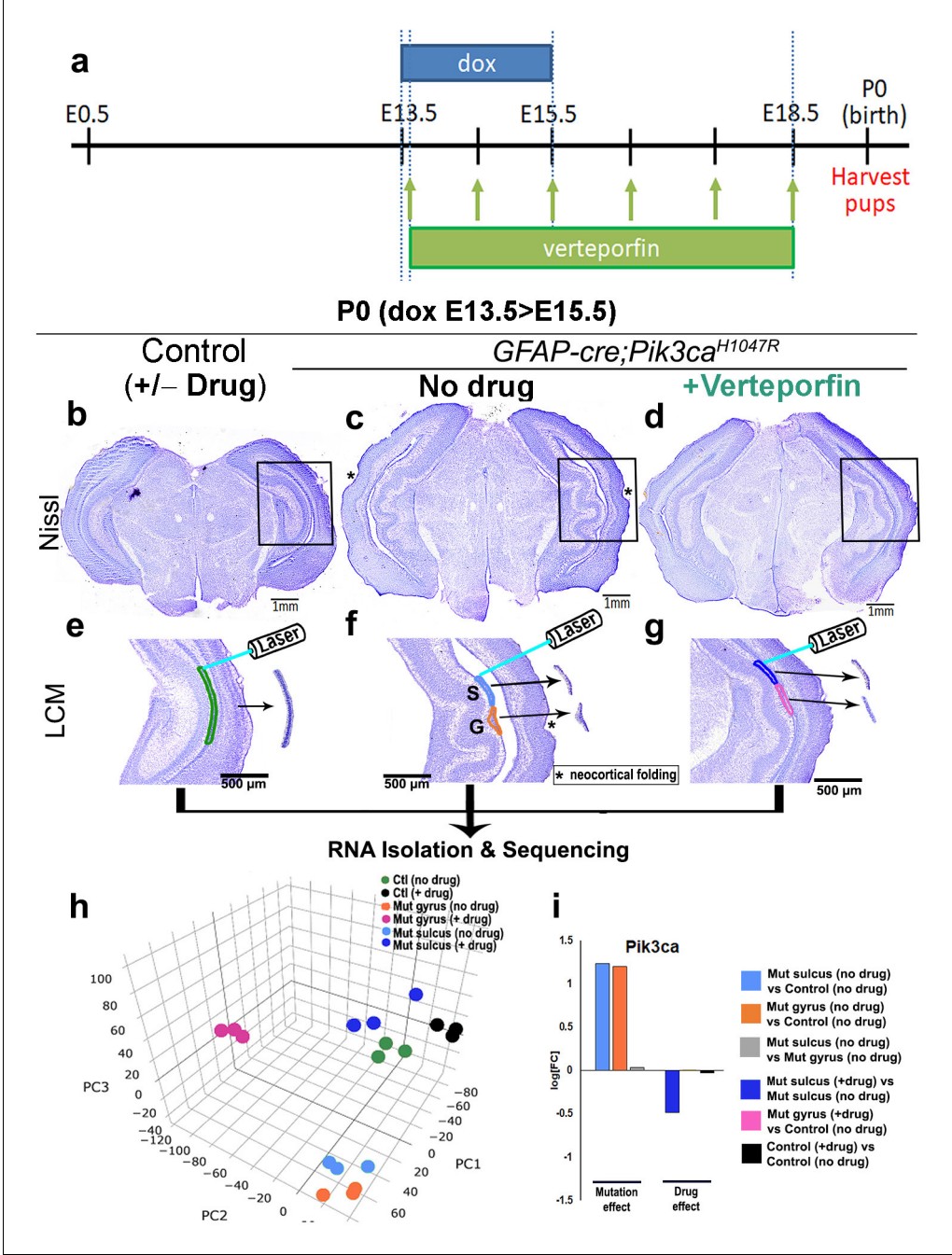

**Figure 6.** Translocation of Yap from nucleus to cytoplasm by verteporfin attenuates gyrification and ventriculomegaly in *Pik3ca*[H1047R] mutant. (**a**) Developmental timeline of verteporfin administration (E13.5 > E18.5). (**b–d**) Nissl-staining revealed attenuation of gyrification and absence of ventriculomegaly in P0 *Pik3ca*[H1047R] mutant (induced E13.5 > E15.5) post-verteporfin administration. Boxes mark the respective areas of interest for (**e–g**). (**e–g**) Flowchart of laser capture microdissection (LCM) of marked ventricular-subventricular zone tissue samples, and RNA sequencing. Asterisks (**c,f**) mark neocortical gyrification in the untreated mutant brains, that were also eventually attenuated by verteporfin (**d,g**). (**h**) Principal component (PC) analysis revealed that verteporfin administration drives ventricular *Pik3ca* mutant gene expression towards that of control tissue. PC1 is largely explained by PI3K signaling, PC2 by Hippo-Yap signaling. (**i**) Graphs showing the effect of PI3K overactivation and of verteporfin treatment in *Pik3ca*. Scale bars: 1 mm (**b–d**), 500 µm (**e–g**). See also *Figure 6—figure supplements 1–3*, *Figure 6—source datas 1* and *2*.

DOI: https://doi.org/10.7554/eLife.45961.016

*Figure 6 continued on next page*

*Figure 6 continued*

The following source data and figure supplements are available for figure 6:

**Source data 1.** Significant gene list and differential expression analysis from P0 *Pik3ca*^H1047R mouse RNA-seq data.
DOI: https://doi.org/10.7554/eLife.45961.020

**Source data 2.** Gene lists used in gene set enrichment analyses.
DOI: https://doi.org/10.7554/eLife.45961.021

**Figure supplement 1.** Verteporfin administration has no overt effect on control littermates.
DOI: https://doi.org/10.7554/eLife.45961.017

**Figure supplement 2.** Barcode plots across groups showing gene enrichment.
DOI: https://doi.org/10.7554/eLife.45961.018

**Figure supplement 3.** *Pik3ca* mutant gyrification is physiologically relevant.
DOI: https://doi.org/10.7554/eLife.45961.019

---

*2h*). The effect of PI3K-Yap interactions on cell adhesion and proliferation is summarized in *Figure 8a–c*.

## Discussion

Cortical gyrification or its absence is an essential feature of mammalian brain evolution but the mechanisms driving cortical folding, particularly its initiation, are poorly understood (*Borrell, 2018*). Using our activating *Pik3ca* mutant mouse model of human cortical malformations (*Roy et al., 2015*), we have established that regulated cell adhesion and proliferation via PI3K-Yap signaling at the apical edge of the embryonic ventricular zone is critical to maintain the lissencephalic nature of mouse brain and prevent developmental hydrocephalus. Overactivation of PI3K signaling in neural progenitors during a critical embryonic period (E13.5-E15.5) led to 'true' gyrification (*Borrell, 2018*) of the mutant mouse neocortex and hippocampus, with the stereotypic characteristics of folded pial surface and neuronal plate/white matter of differential thickness, with predominantly smooth apical ventricular surfaces. This transient PI3K activation was also sufficient to initiate a stereotypic neurogenic sequence similar to that seen in naturally gyrencephalic mammals, such as ferrets (*Reillo and Borrell, 2012*; *de Juan Romero et al., 2015*). This sequence includes differential proliferation of primary and secondary (IPs, bRGs) progenitors, cell differentiation and migration, and an altered radial glial scaffold. PI3K overactivation did not disrupt gross hippocampal patterning but did cause subtle disruptions in cell adhesion at the apical edge. Adhesion abnormalities were concurrently associated with aberrant focal cytoplasmic-to-nuclear translocation of Yap protein, which predicted the stereotypical positioning of the gyral foci. Hydrocephalus in this model was also associated with the disruption of PI3K-dependent apical adhesion in the same progenitors, but across a broader embryonic time-period. Both the gyrification and hydrocephalic phenotypes were 100% penetrant in the mutants and were attenuated by embryonic treatment with verteporfin, a nYap inhibitor, which normalized apical junctions and proliferation along the mutant ventricular linings. Our data provide new insights regarding the mechanisms that initiate gyrification in mammals, and apical adhesion alterations in human PI3K-related disorders that contribute to the broad range of neuropathology including impaired brain folds and ventriculomegaly.

### *Pik3ca* mis-regulation altered cell adhesion at the apical edge of cortical ventricular zone

Brief activation of the *Pik3ca*^H1047R mutation resulted in focal disruption of adhesion, as observed in the loose 'zippering' of the ventricular linings, subtle initial changes and then gaps in the expression of adhesion molecules β-Catenin, N-Cadherin and ZO-1, especially in the prospective sulci. This in turn impinged on the development of neuronal and ependymal progenitors. Interestingly, the developing ferret brain also demonstrates differential regional variation in cell adhesion and junctional proteins at the prospective gyrus and sulcus (*de Juan Romero et al., 2015*). Genes that were differentially expressed between the gyrus and sulcus in ferret were enriched in our *Pik3ca*^H1047R mutant, with considerable overlap in cell adhesion-related genes. This demonstrates potential physiological relevance of our model to naturally gyrencephalic mammals, although additional experiments may be required.

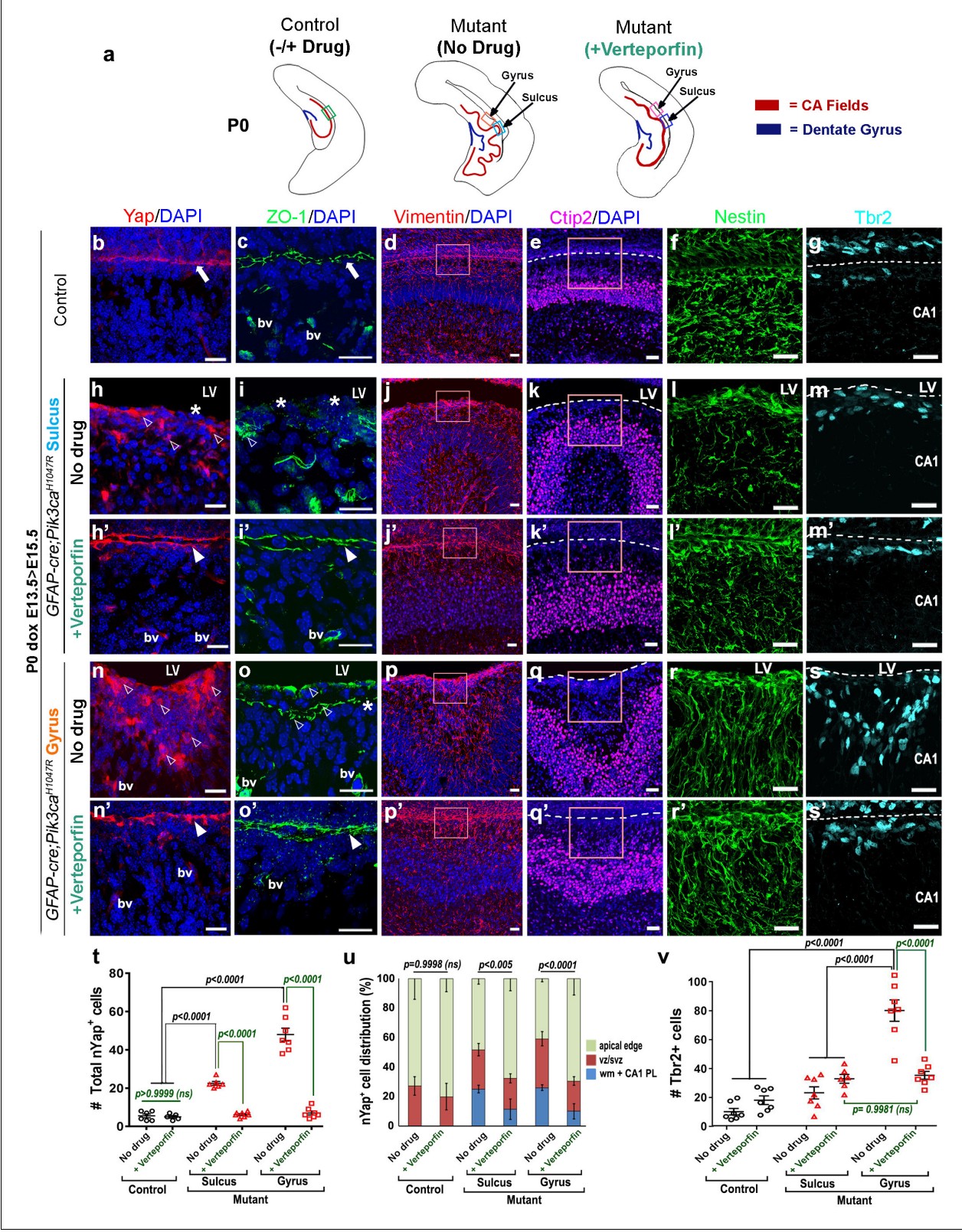

**Figure 7.** Reduction of nYap by verteporfin re-establishes developing apical junctions and suppresses focal proliferation in *Pik3ca^{H1047R}* mutant. (**a**) Schematics of P0 control and mutant (±verteporfin) forebrain hemi-sections; boxes depict respective areas of interest. (**b–g**) P0 control untreated brain expressed cYap, adherens junction protein ZO-1 and radial glial markers Vimentin and Nestin, uniformly along the juxtaposed apical membranes. Ctip2 and Tbr2 were expressed in pyramidal neurons and IPs respectively. (**h–s'**) Verteporfin treatment significantly normalized Yap localization, from nucleus

*Figure 7 continued on next page*

*Figure 7 continued*

to cytoplasm, restored apical junctions and streamlined disrupted radial glial scaffold, as compared to the untreated P0 (dox E13.5 > E15.5) mutant gyri and sulci. (t,u) The treatment significantly reduced the total nYap+ cell number in the mutant hippocampal gyri and sulci (F = 70.47, df = 32) as well as normalized the nYap+ cell distribution to the control levels (F = 134.4, df = 96). This also significantly decreased the Tbr2+ progenitor population (F = 31.45, df = 35), especially in the mutant gyral zones, as quantitated in (v). This resulted in attenuation of the extent of gyrification and re-zippering of the apical membranes to halt the progressive ventriculomegaly. Data are represented as mean ± SEM in scatter plots (t); two-way ANOVA was followed by Tukey's post-tests. ns, not significant. Open arrowheads, disrupted/ectopic junctional and ependymal proteins in mutant; asterisks, absence of adhesion/ependymal molecules at the mutant apical edge; arrowheads, rescued apical ventricular lining in the mutant post-verteporfin treatment; bv, blood vessels; LV, lateral ventricle. Scalebars: 20 μm (b–s'). See also *Figure 7—figure supplements 1–2*.

DOI: https://doi.org/10.7554/eLife.45961.022

The following figure supplements are available for figure 7:

**Figure supplement 1.** Reduction of nYap by verteporfin re-establishes cell adhesion in *Pik3ca^H1047R* mutant.
DOI: https://doi.org/10.7554/eLife.45961.023

**Figure supplement 2.** Effect of verteporfin on apical junctions initiates early.
DOI: https://doi.org/10.7554/eLife.45961.024

The ventricular surface of the developing brain is formed by tangential assembly of the apical end-feet of progenitor cells, enriched in adherens junction molecules, ZO-1 and Cadherins (*Nagasaka et al., 2016*). Loss-of-function mutations in the cell adhesion machinery, such as *fyn*,

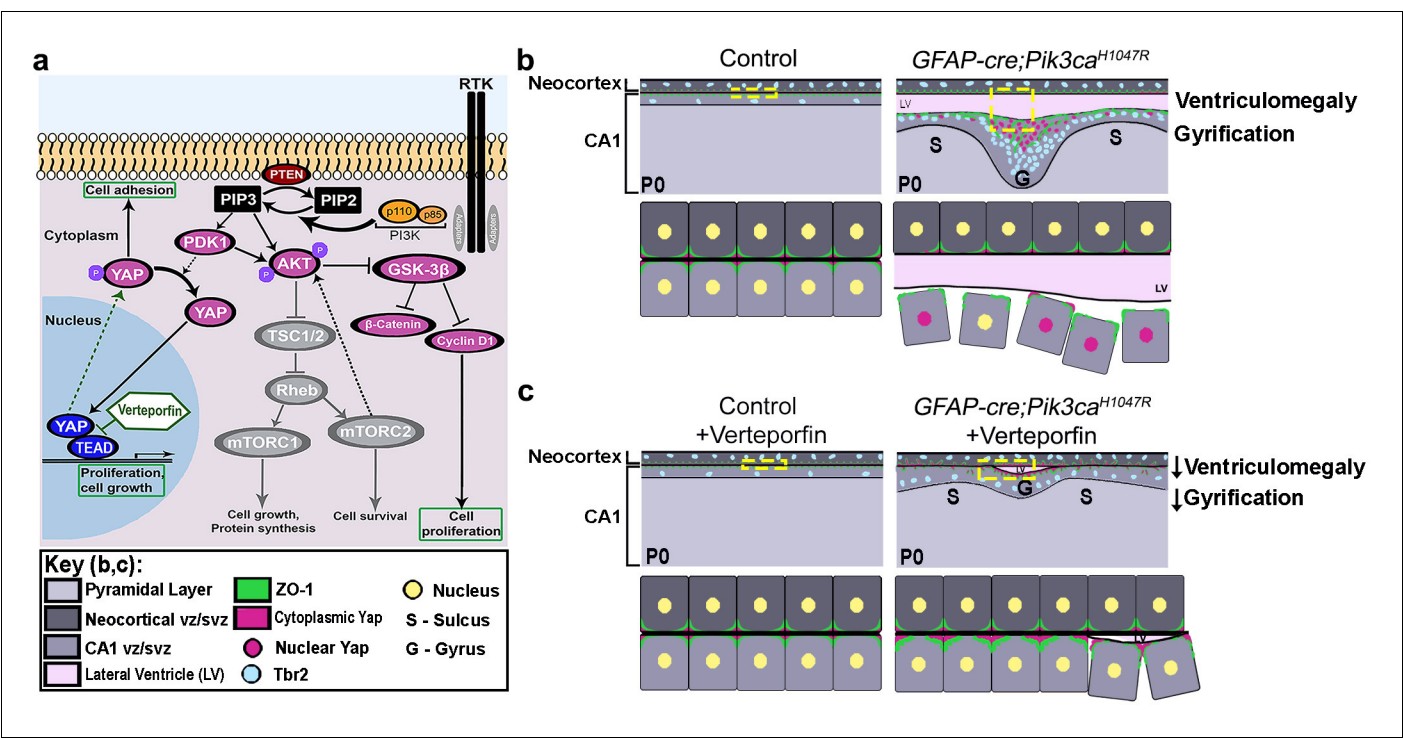

**Figure 8.** Summary: PI3K-Yap gyrification model. (a) Schematic of PI3K-AKT-MTOR pathway, its connection with Yap signaling and mode of verteporfin action. The activity of the PI3K enzyme (dimer of catalytic and regulatory subunits) initiates the PI3K-AKT-MTOR pathway cascade. In parallel, downstream of the Hippo pathway, Yap binds to transcriptional regulator proteins in the nucleus, to facilitate cell growth and proliferation. Activation of Hippo pathway results in translocation of nuclear Yap to cytoplasm through phosphorylation. Cytoplasmic Yap, in turn, promotes cell adhesion. The activating *Pik3ca* mutation results in Yap translocation from cytoplasm to nucleus, possibly via Pdk1. In this study, verteporfin, a nuclear Yap inhibitor, restored cytoplasmic Yap. (b,c) Schematics summarizing main findings of the paper: PI3K over-activation resulted in disruption of cell adhesion at the neural-ependymal transition zone causing ventriculomegaly, and also in differential proliferation of progenitors, thus triggering the gyrification sequence cascade (effect of genotype). In mutant mice, verteporfin attenuated these anomalies (effect of drug), leading to reduction in gyrification severity as well as ventriculomegaly. Area within the yellow dashed boxes in the top rows of (b) and (c) is magnified in the respective bottom rows. Color coding for elements in (b) and (c) is explained in the figure 'key'. vz/svz, ventricular/subventricular zone.

DOI: https://doi.org/10.7554/eLife.45961.025

*Cdh2* (N-Cadherin) cause dysplastic hippocampal morphology (*Grant et al., 1992*; *Kadowaki et al., 2007*). Recently, adhesion-ECM molecules, like E-cadherin, connexins, FLRT1/3, filamin, have been identified, that influence the balance in adhesion-repulsion forces between migrating neurons to cause interspersed cell clustering with differential migration speed, eventually establishing the cortical sulci (*Borrell, 2018*; *Llinares-Benadero and Borrell, 2019*). It is interesting to note that all these molecules have well-established functional interactions with the PI3K pathway (*Gout et al., 1992*; *Tran et al., 2002*; *Najib et al., 2012*; *Huang et al., 2015*; *Martinez et al., 2015*).

Perhaps the closest published model to our new *Pik3ca^{H1047R}* mouse model is the *D6-cre;Cdh2* conditional knockout (*Kadowaki et al., 2007*), which exhibited folding in the CA1 PL. However, the loss of N-Cadherin in early differentiating neurons induced complete disruption of adherens junctions, resulting in abnormalities of internal cortical structures, dysplastic radial glial architecture and cellular lamination, and blurring of the lateral neocortex–hippocampus delineation. Such severe developmental defects are neither observed in normal gyrencephalic mammals nor in our mutant mouse model. Rather in our model, Pik3ca activation caused much more subtle disruption of apical junctions, resulting in cortical folding with broadly normal lamination, as observed in naturally gyrencephalic animals.

The PI3K enzyme is composed of p110 (encoded by *Pik3ca*) and p85 subunits (*Geering et al., 2007*), Previous studies have demonstrated that *Pik3ca^{H1047R}* allele remains tightly dependent on p85 regulation (*Geering et al., 2007*; *Zhao and Vogt, 2008*; *Roy et al., 2015*). Hence, although the *Pik3ca^{H1047R}* allele is a strongly activating mutation with higher kinase activity (*Gymnopoulos et al., 2007*), it is functionally not an overexpression allele to cause massive pathway induction. Taken together, our data indicate that exquisite PI3K-dependent regulation of apical cell adhesion is essential for normal mouse brain development and suggests that subtle regional changes in other mammals may initiate natural gyrification. This hypothesis should be readily testable but is beyond the scope of our current study.

## PI3K-dependent modulation of Yap protein function is central to initiation of gyrification

Focal loss of apical integrity and over-proliferation of neural progenitors in our *Pik3ca* model highly correlated with focal areas of translocation of Yap protein from the apical cytoplasmic edge to nuclei at the ventricular lining. Yap is a central regulator of the Hippo pathway, with both nuclear and cytoplasmic functions (*Gumbiner and Kim, 2014*; *Park et al., 2016*; *Park et al., 2018*). The Hippo-Yap pathway is known to mediate mechanical signals, cellular stress, polarity and adhesion cues that are integrated through multiple upstream regulators (*Hansen et al., 2015*). Phosphorylated Yap is sequestered in the cytoplasm by adherens junction proteins. In turn, cYap, expressed in the early developing ependyma, has been shown to critically maintain cell-cell adhesion at the ventricular edge of ventral aqueduct (*Park et al., 2016*). When dephosphorylated, Yap translocates to the nucleus, where it binds to TEAD transcriptional co-regulators to modulate gene expression (*Hansen et al., 2015*; *Park et al., 2018*). In this context, nYap was recently shown to be sufficient to drive cortical progenitor proliferation (*Lavado et al., 2018*).

We hypothesized that PI3K regulation of Yap subcellular distribution was central to the gyrification phenotype in our *Pik3ca^{H1047R}* mutant mouse model. *In vitro*, PI3K-PDK1 pathway activation promotes nuclear translocation of Yap, thus disrupting contact inhibition (*Fan et al., 2013*). Verteporfin, a nYap antagonist, disintegrates the Yap-TEAD nuclear complex, facilitating sequestration of Yap to the cytoplasm (*US Food and Drug Administration, 2000*; *Schmidt-Erfurth and Hasan, 2000*; *Liu-Chittenden et al., 2012*; *Fan et al., 2013*; *Brodowska et al., 2014*; *Gumbiner and Kim, 2014*). Treatment of *Pik3ca^{H1047R}* mutants with verteporfin restored the apical cYAP distribution in progenitors, rescued the integrity of the ventricular surface, reduced the enhanced progenitor proliferation and subsequently attenuated the gyrification of the *Pik3ca^{H1047R}* mutant mice. To our knowledge, this is the first study where verteporfin was introduced intraperitoneally into the pregnant mice to successfully suppress nYap function in embryos across the placental barrier.

We did not fully rescue the gyrification phenotype, likely because PI3K signaling has a multitude of downstream outputs in addition to Yap regulation. Although we cannot rule out potential off-target effects of verteporfin, our data provide strong evidence that PI3K pathway modulation of the cYap pool during the brief embryonic critical period is essential to maintain the lissencephalic mouse brain. We speculate that subtle changes in local apical adhesion properties of early progenitors,

associated with focal changes in Yap subcellular distribution and subsequently in the basal progenitor population, is one of the initiating mechanisms of gyrification in higher mammals. This may also be linked to changes in centrosome and microtubule organization in apical progenitors (*Camargo Ortega et al., 2019*). Further, a recent study has demonstrated that YAP maintains basal progenitor population in the developing ferret and human neocortex (*Kostic et al., 2019*). Similar to our current findings, this study also showed that YAP regulation by verteporfin in *ex vivo* embryonic ferret and fetal human neocortical free-floating tissue culture systems leads to reduction in basal progenitor number (*Kostic et al., 2019*). However, further analysis of all these parameters in ferrets (*Borrell, 2018*) and other model systems is required.

Cortical folding patterns in naturally gyrencephalic species are highly stereotypic, although there is considerable inter-individual variation (*Ronan and Fletcher, 2015*; *Borrell, 2018*). Importantly, the gyrification pattern induced in *GFAP-cre;Pik3ca$^{H1047R}$* mutants was non-random, with 100% penetrance, and was highly cre-dependent (*Roy et al., 2015*; *D'Gama et al., 2017*). We do not currently understand why gyri form in stereotypical positions in *Pik3ca$^{H1047R}$* mutant mice or indeed, in any normal gyrencephalic mammal. We hypothesize that cell-cell adhesion is less constrained in susceptible focal positions compared to adjacent regions during the critical periods of development. These coordinates may be defined by genetically encoded signaling gradients or may simply be a result of regional variations in mechanical force (*Ronan and Fletcher, 2015*; *Borrell, 2018*). Developing cortical tissue from naturally gyrencephalic mammals is of limited availability, making extensive experimentation difficult. The highly regulatable stereotypical and fully penetrant gyrification phenotype of *Pik3ca$^{H1047R}$* mutant mice provides a new, highly tractable alternative model to unravel the positional constraints giving rise to brain folds.

### Yap-dependent apical cell adhesion mediates normal ependymal development and its disruption likely underlies multiple types of pediatric hydrocephalus amenable to small molecule therapy

In addition to gyrification, embryonic induction of *Pik3ca$^{H1047R}$* allele in mouse neural progenitors also caused ventriculomegaly which progressed postnatally to hydrocephalus. Hydrocephalus is a common feature of human PI3K-related brain overgrowth syndromes (*Mirzaa et al., 2012*; *Tully and Dobyns, 2014*; *Guerra et al., 2015*; *Khan et al., 2015*). We determined that PI3K-dependent ventriculomegaly had long embryonic critical period, ending at E17.5, compared to the short gyrification critical period (E13.5-E15.5). Embryonic verteporfin-dependent nYap inhibition also rescued ventriculomegaly, with minimal effect on control littermates. We conclude that disrupted apical cell adhesion and aberrant nYap localization were central to this phenotype as well as gyrification.

Hydrocephalus is often associated with a disrupted ependyma, although in many cases it is unclear if this is because of abnormal ependymal development or ependymal damage at a later stage (*Jiménez et al., 2014*). The ependyma is a continuous single layer of neuroepithelial multi-ciliated cells lining the lateral ventricles. Ependymal cells differentiate from apical progenitors and mature during late embryogenesis and early postnatal mouse development, when most definitive ependymal markers, including FoxJ1 and other markers of multi-ciliated cells, are expressed (*Schmidt-Erfurth and Hasan, 2000*; *Lavado and Oliver, 2011*). Little is understood regarding the primary molecular control of ependymal specification. In our mutant mouse model, the normally uniform layer of Vimentin$^+$/Yap$^+$ developing ependymal cells was disrupted at mid-embryonic stages. Further, cells with ependymal fate (Yap$^+$) were intermingled with Tbr2$^+$ progenitors and young Ctip2$^+$ neurons at the mutant gyral edge, indicating abnormal neural-to-ependymal transition. This phenotype and the subsequent hydrocephalus were reversed with verteporfin treatment, although the drug-treated ependyma was not entirely normal. These data clearly demonstrate that early Yap-dependent apical adhesion is required for normal ependymal development in forebrain during embryonic pre-ciliogenesis stages – a developmental time much earlier than previously established.

Interestingly, human post-mortem studies and *in vitro* model studies have shown disruption of cell adhesion and apical junctions in post-hemorrhagic hydrocephalus, a common form of hydrocephalus in premature infants, currently with few treatment options (*Morales et al., 2012*; *Jiménez et al., 2014*; *Guerra et al., 2015*). In mice, embryonic intraventricular administration of blood-derived lysophosphatidic acid (LPA), is sufficient to cause disrupted ependymal cell adhesion and neonatal hydrocephalus in mice (*Yung et al., 2015*). More recently, *Park et al. (2016)* showed that depletion of cYAP, by either genetic deletion or LPA treatment, also resulted in perinatal

hydrocephalus due to impaired ependymal development and aqueduct blockage (*Park et al., 2016*). Taken together with our data, we posit that altered periventricular cell adhesion caused by dysregulated Hippo-Yap signaling is a common convergent mechanism for both developmental and post hemorrhagic hydrocephalus. Fine regulation of the Hippo-Yap pathway may represent a new therapeutic approach for pediatric hydrocephalus patients, especially since verteporfin effectively crosses the placenta and has minimal effect on normal brain development as demonstrated by our morphological and RNA-seq analyses. This implication is worthy of further detailed preclinical analysis.

In summary, building upon our original study of mouse models of human *PIK3CA*-related brain malformations (*Roy et al., 2015*), we demonstrate intimate PI3K-dependent developmental and molecular links between cortical neurogenesis and ependymal development at the apical edge of the embryonic ventricular zone. Disruption of apical surface integrity in the forebrain, via enhanced embryonic PI3K-Yap signaling, alters neurogenesis and can initiate *bona fide* gyrification in mice. This process mimics the canonical neurogenic sequence observed in naturally gyrencephalic mammals. Concurrently, abnormal apical PI3K-Yap interaction disrupts ependymal development in forebrain, prior to ciliogenesis, leading to hydrocephalus. Our data readily explain the coincidence of impaired neurogenesis, gyrification and hydrocephalus, observed often in patients with cortical dysplasia and in our *Pik3ca* mouse model (*Keppler-Noreuil et al., 2014*; *Jansen et al., 2015*; *Roy et al., 2015*; *Parrini et al., 2016*; *Furey et al., 2018*). The results also provide support for nuclear Yap protein as a potential new therapeutic target for these clinically important disorders.

# Materials and methods

## Key resources table

| Reagent type (species) or resource | Designation | Source or reference | Identifiers | Additional information |
|---|---|---|---|---|
| Genetic reagent (*Mus musculus*) | *GFAP-cre* | gift (JJ Zhao); PMID: 11668683 | (IMSR Cat# JAX:004600, RRID:IMSR_JAX:004600) | gifted by Dr. Jean J Zhao (Dana Farber Cancer Inst., Boston, USA) |
| Genetic reagent (*Mus musculus*) | *Pik3ca^{H1047R}* | gift (JJ Zhao); PMID: 21822287 | MGI: 5526971 | gifted by Dr. Jean J Zhao (Dana Farber Cancer Inst., Boston, USA) |
| Genetic reagent (*Mus musculus*) | *Ai14/+; Ai14* | Jax labs (stock #007914), PMID: 20023653 | (IMSR Cat# JAX:007914, RRID:IMSR_JAX:007914) | PMID: 20023653 |
| Genetic reagent (*Mus musculus*) | *Rosa26-rtTA; Rosa* | PMID: 15784609 | (IMSR Cat# JAX:005670, RRID:IMSR_JAX:005670) | gifted by Dr. Jean J Zhao (Dana Farber Cancer Inst., Boston, USA) |
| Antibody | Anti-BrdU antibody [BU1/75 (ICR1)] | Abcam | (Abcam Cat# ab6326, RRID:AB_305426) | IHC (1:100) |
| Antibody | Anti-BrdU-Fluorescein antibody Formalin grade | Roche | (Roche Cat# 11202693001, RRID:AB_514484) | IHC (1:100) |
| Antibody | Rabbit anti Calbindin D-28k | Swant | (Swant Cat# CB38, RRID:AB_2721225) | IHC (1:1500) |
| Antibody | Anti-Ctip2 antibody [25B6] - ChIP Grade | Abcam | Abcam Cat# ab18465, RRID:AB_2064130) | IHC (1:250) |
| Antibody | Rabbit GFAP antibody | Dako (now Agilent) | (Agilent Cat# Z0334, RRID:AB_10013382) | IHC (1:2500) |

*Continued on next page*

Continued

| Reagent type (species) or resource | Designation | Source or reference | Identifiers | Additional information |
|---|---|---|---|---|
| Antibody | Anti-Neural Cell Adhesion Molecule L1 Antibody, clone 324 | Millipore | (Millipore Cat# MAB5272, RRID:AB_2133200) | IHC (1:200) |
| Antibody | Rabbit Anti-Laminin | Sigma | (Sigma-Aldrich Cat# L9393, RRID:AB_477163) | IHC (1:25) |
| Antibody | Purified Mouse Anti-N-Cadherin | BD Biosciences | (BD Biosciences Cat# 610921, RRID:AB_398236) | IHC (1:150) |
| Antibody | Mouse Anti-Nestin Antibody, clone rat-401 | Millipore | (Millipore Cat# MAB353, RRID:AB_94911) | IHC (1:200) |
| Antibody | Rabbit Purified anti-Pax-6 Antibody | Biolegend | (BioLegend Cat# 901301, RRID:AB_2565003) | IHC (1:300) |
| Antibody | Mouse anti-Pax6 antibody | DSHB | N/A | IHC (1:300; deposited in DSHB by Kawakami A.) |
| Antibody | Phospho-Histone H3 (Ser10) (6G3) Mouse mAb | Cell Signaling | (Cell Signaling Technology Cat# 9706, RRID:AB_331748) | IHC (1:200) |
| Antibody | Anti-Reelin Antibody, a.a. 164–496 mouse reelin, clone G10 | Millipore | (Millipore Cat# MAB5364, RRID:AB_2179313) | IHC (1:1200) |
| Antibody | Anti-SATB2 antibody [SATBA4B10] - C-terminal | Abcam | (Abcam Cat# ab51502, RRID:AB_882455) | IHC (1:400) |
| Antibody | Rabbit anti-Six3 polyclonal antibody; | Rockland Antibodies | (Rockland Cat# 600–401-A26S, RRID:AB_11181864) | IHC (1:200) |
| Antibody | Rabbit anti-Sox2 antibody | Thermo Fisher Scientific | (Thermo Fisher Scientific Cat# PA1-094, RRID:AB_2539862) | IHC (1:400) |
| Antibody | Rabbit anti-Tbr1 antibody | Millipore | (Millipore Cat# AB10554, RRID:AB_10806888) | IHC (1:400) |
| Antibody | EOMES Monoclonal Antibody (Dan11mag) Tbr2 antibody | eBioscience | (Thermo Fisher Scientific Cat# 14-4875-82, RRID:AB_11042577) | IHC (1:200) |
| Antibody | Anti-Vimentin antibody [EPR3776] - Cytoskeleton Marker | Abcam | (Abcam Cat# ab92547, RRID:AB_10562134) | IHC (1:200) |
| Antibody | YAP (D8H1X) XP Rabbit mAb | Cell Signaling | (Cell Signaling Technology Cat# 14074, RRID:AB_2650491) | IHC (1:100) |
| Antibody | ZO-1 Monoclonal Antibody (ZO1-1A12), Alexa Fluor 488 | Thermo Fisher Scientific | (Thermo Fisher Scientific Cat# 339188, RRID:AB_2532187) | IHC (1:500) |

*Continued*

| Reagent type (species) or resource | Designation | Source or reference | Identifiers | Additional information |
|---|---|---|---|---|
| Antibody | Purified Mouse Anti-β-Catenin Clone 14 | BD Biosciences | (BD Biosciences Cat# 610154, RRID:AB_397555) | IHC (1:200) |
| Antibody | Rabbit β-Catenin antibody | Abcam | (Abcam Cat# ab2365, RRID:AB_303014) | IHC (1:100) |
| Antibody | Goat anti-Rat IgG (H + L) Cross-Adsorbed Secondary Antibody, Alexa Fluor 647 | Thermo Fisher Scientific | (Thermo Fisher Scientific Cat# A-21247, RRID:AB_141778) | IHC (1:400) |
| Antibody | Goat anti-Mouse IgG (H + L) Highly Cross-Adsorbed Secondary Antibody, Alexa Fluor 488 | Thermo Fisher Scientific | (Thermo Fisher Scientific Cat# A-11029, RRID:AB_2534088) | IHC (1:400) |
| Antibody | Goat anti-Rabbit IgG (H + L) Highly Cross-Adsorbed Secondary Antibody, Alexa Fluor 568 | Thermo Fisher Scientific | (Thermo Fisher Scientific Cat# A-11036, RRID:AB_10563566) | IHC (1:400) |
| other | DAPI stain | Molecular Probes | (Thermo Fisher Scientific Cat# D1306, RRID:AB_2629482) | 1:10000 |
| Commercial assay or kit | ApopTag Plus Fluorescein In situ Apoptosis Detection Kit | Chemicon/Millipore | Cat# S7111 | |
| Commercial assay or kit | SMART-Seq v4 Ultra Low Input RNA Kit | Takara | Cat# 634889 | |
| Commercial assay or kit | Nextera XT kit | Illumina | Cat# FC-131–1024 | |
| Commercial assay or kit | KAPA qPCR complete kit | Biorad | KK4844 Complete kit 500 × 20 ul reactions | |
| Commercial assay or kit | RNeasy Micro Kit | Qiagen | Cat# 74004 | |
| Commercial assay or kit | Bioanalyzer 6000 Pico Kit | Agilent | Cat# 5067–1513 | |
| Chemical compound, drug | Verteporfin | US Pharmacopeial Convention | Cat# 1711461 | |
| Chemical compound, drug | BrdU Labeling Reagent | Life Technologies | Cat# 000103 | |
| Chemical compound, drug | Doxycycline (doxycycline hyclate) | Sigma | Cat# D9891-25G | |
| Software, algorithm | BioViz3D version 3.1 | BioViz3D | BioViz3D, RRID:SCR_017162 | |
| Software, algorithm | salmon aligner v0.11.3 for R | PMID: 28263959 | | |
| Software, algorithm | Bioconductor tximport package, v1.8.0 for R | PMID: 26925227 | | |
| Software, algorithm | Bioconductor edgeR package v3.22.3 | PMID: 19910308 | (Bioconductor, RRID:SCR_006442) | Software |

*Continued on next page*

*Continued*

| Reagent type (species) or resource | Designation | Source or reference | Identifiers | Additional information |
|---|---|---|---|---|
| algorithm | glmTreat function in edgeR | PMID: 19176553 | | |
| Software, algorithm | Bioconductor GEOquery v2.48.0 | PMID: 17496320 | (GEOquery, RRID:SCR_000146) | |
| Software, algorithm | limma packages, v3.36.2 for R | PMID: 25605792 | | |

## Mice

The following mouse lines were used: human glial fibrillary acidic protein (*hGFAP*)-*cre* (Jackson Labs, IMSR Cat# JAX:004600, RRID:IMSR_JAX:004600) (*Zhuo et al., 2001*), mentioned as *GFAP-cre* in this study; *Pik3ca^{H1047R}* transgenic (human *Pik3ca^{H1047R}* transgene expression is under the control of a tetracycline-inducible promoter (TetO)), *Rosa26-rtTA* line (Jackson Labs, Stock #005670) (*Belteki et al., 2005*), Ai-14 (Jackson Labs, Stock #007914) (*Madisen et al., 2010*).

All lines were maintained on a mixed genetic background, comprising of FVB, 129 and CD1 strains. All mice were housed in Optimice cages with aspen bedding at the Seattle Children's Research Institute's specific pathogen-free (SPF) vivarium facility. Noon of the day of vaginal plug was designated as embryonic day 0.5 (E0.5). The day of birth was designated as postnatal day 0 (P0). The *Pik3ca^{H1047R}* and *Rosa26-rtTA* lines were intercrossed and female mice positive for both these alleles were crossed with *GFAP-cre;RosartTA; Pik3ca^{H1047R}* males. Embryos or pups of both sexes, genotyped positive for all three genes, namely *cre, Rosa, Pik3ca^{H1047R}*, were used in this study. The *GFAP-cre* driver gets activated in neural progenitors only at ~E13 (*Zhuo et al., 2001*; *Roy et al., 2015*). The activating *H1047R* mutation in the PI3K catalytic subunit (*Pik3ca*) increases the level and duration of response to extracellular ligands, its stability being completely dependent on the unaltered level of the p85 regulatory subunit (*Roy et al., 2015*). To ensure that *cre* and *Pik3ca^{H1047R}* mutant transgene expression was correlated, plugged females were treated with doxycycline (Sigma; 2 mg/ml) from E0.5 or as mentioned in the text, available *ad libitum* in drinking water. For the neonatal induction experiment, the pups were treated with doxycycline from P1. The *GFAP-cre;RosartTA;Pik3ca^{H1047R}* mutant is mentioned as *Pik3ca^{H1047R}* mutant in the text. Mouse genotyping by PCR was done using separate sets of primers for the *Cre* coding region, and the *Pik3ca^{H1047R}*, *Rosa+/-*, and *Ai14/+* alleles, as previously described (*Roy et al., 2015*). All mouse procedures were approved and conducted in accordance with the guidelines laid down by the Institutional Animal Care and Use Committees (IACUC) of Seattle Children's Research Institute, Seattle, WA, USA.

## Sample preparation and histochemical procedures

Embryos and postnatal pups were harvested in phosphate buffer saline (PBS); brains fixed in 4% paraformaldehyde (PFA) for 4 hr, equilibrated in 30% (wt/vol) sucrose made in PBS, and sectioned at 25 μm on a freezing microtome. Sections were then processed for Nissl or immunohistochemical staining. No data were excluded from our analysis and no randomization was used. Tissue collection was not performed blind since the mice were subjected to genotyping and drug administration. However, the data analysis was performed blinded by at least two individuals.

### Immunohistochemistry (IHC)

Sections were washed thrice in PBS, boiled in 10 mM Sodium citrate solution for antigen retrieval, blocked in 5% serum in PBS with 0.1% Triton X-100 and then incubated overnight at 4°C with primary antibodies. The next day, sections were washed thrice in PBS, incubated with appropriate species-specific secondary antibodies conjugated with Alexa 488, 568 or 647 fluorophores (Invitrogen) for 2 hr at room temperature and then counterstained with DAPI (4',6-Diamidino-2-Phenylindole, Dihydrochloride; Invitrogen; D1306) to visualize nuclei. Sections were cover-slipped using Fluorogel (Electron Microscopy Sciences, EMS #17985) mounting medium. Immuno-stained sections were imaged in Zeiss LSM 710 Imager Z2 laser scanning confocal microscope using Zen 2009 software

and in Olympus VS-120 slide-scanner microscope using Olympus VS-Desktop 2.9 software, and later processed in ImageJ 1.51j8 and ImageJ2 (NIH, Bethesda, Maryland, USA) and Olympus VS-Olyvia 2.9 software programs respectively. Each antibody was validated for mouse and application (IF, IHC) by the correspondent manufacturer, and is publicly available on its website with indicated catalog numbers. This was also validated by us in our experiments, replicating published/expected expression in control tissue. Primary antibodies used: rat anti-BrdU (Abcam), mouse anti-BrdU (Roche), rabbit anti-Calbindin (Swant), rat anti-Ctip2 (Abcam), rabbit anti-GFAP (Dako), rat anti-L1 (Millipore), rabbit anti-Laminin (Sigma), mouse anti-N-Cadherin (BD Biosciences), mouse anti-Nestin (Millipore), rabbit anti-Pax6 (Biolegend), mouse anti-Pax6 (DSHB), mouse anti-phospho-Histone H3 (Cell Signaling), mouse anti-Reelin (Millipore), mouse anti-Satb2 (Abcam), rabbit anti-Six3 (Rockland Antibodies), rabbit anti-Sox2 (Thermofisher), rabbit anti-Tbr1 (Millipore), rat anti-Tbr2 (eBioscience), rabbit anti-Vimentin (Abcam), rabbit anti-Yap (Cell Signaling), mouse anti-ZO-1 (), mouse anti-β-Catenin (BD Biosciences), rabbit anti-β-Catenin (Abcam). Immuno-histochemistry replication consisted of performing the same experiment with biologically independent samples from the same group (control or mutant ±drug). All attempts for replication were successful. No outliers were encountered. Each antibody was validated for mouse and application (IHC) by the correspondent manufacturer and is publicly available on its website with indicated catalog numbers. This was also validated by us in our experiments, replicating published/expected expression in control tissue.

## Nissl staining

Sections were stained in 0.1% cresyl violet solution (Cresyl violet, Sigma Cat# C5042) for 10 min, rinsed quickly in distilled water, dehydrated in 95% ethanol, and left in xylene before being coverslipped with Permount (Thermo Fischer Scientific, SP15-500). Brightfield images were taken in the Olympus VS-120 slide-scanner with Hamamatsu digital camera C11440 and processed using the Olympus VS-Olyvia 2.9 software.

## BrdU incorporation experiments

Bromodeoxyuridine (BrdU; Life Technologies) was administered intraperitoneally (100 µg/g of body weight) to pregnant dams at E14.5/16.5 for 1 hr, at E15.5 for 1 day and at E16.5 for proliferation assays, cell cycle exit and birth-dating experiments respectively. Labeling index and quit fraction were calculated as previously described (*Roy et al., 2015*).

## 3-D modeling and video

3-D models of P3 (dox E0.5 to P3) control and *GFAP-cre;Pik3ca*$^{H1047R}$ mutant hippocampus were developed using the software BioViz3D version 3.1. About 16 serial coronal sections (30 microns each) of Nissl-stained P3 control and mutant hemi-sections of forebrain were imaged and arranged in the rostro-caudal order. Contours were drawn in each section, based on morphology for the entire medial forebrain tissue and the hippocampus. The 3-D reconstruction was created using these contours from each section. The control and mutant models were marked purple and red respectively. The medial tissue for both the control and mutant were marked in a lighter shade of the afore mentioned color, to be distinguished from the darker PL. The movie was made using the movie maker software within BioViz3D and edited with VLC media player.

## Verteporfin treatment

Verteporfin (20 mg/ml stock solution made in 100% mineral oil) was administered intraperitoneally at the dose of 1 µl/g of body weight, one injection per day into timed pregnant dams from E13.5 to E18.5, and the neonatal pups were harvested at P0. The number of mouse brains analyzed was 7/ genotype (±verteporfin), obtained across 5 P0 litters. 100% of the mutant brains demonstrated attenuation of the gyrification and hydrocephalus phenotypes post-verteporfin treatment. For the shorter experiment, same dose of verteporfin was administered to pregnant dams from E13.5 to E16.5 and embryos were harvested at E16.5 (n = 4 brains/genotype/condition).

## TUNEL staining

TUNEL staining was processed on E16.5 control and mutant sections (±Verteporfin) using ApopTag Plus Fluorescein In situ Apoptosis Detection Kit.

## Laser capture micro-dissection (LCM)

Whole forebrains were dissected from P0 (dox E13.5 > E15.5) control and mutant mice (±verteporfin). These intact forebrains were then embedded in OCT, frozen at −80°C, and cryo-sectioned at 16 µm in the coronal plane onto PEN Membrane Glass Slides (Leica Microsystems, USA). The sections were stained with Cresyl Violet (Nissl stain). LCM was performed using the Leica LMD-6000 Laser Microdissection system to capture tissue containing the hippocampal CA1 ventricular lining of control brains and mutant hippocampal gyri and sulci, from each of the 12–14 sections mounted per slide into collection tubes. Total RNA was isolated from LCM-enriched samples pooled across ~6 slides per genotype using the Qiagen RNeasy Micro Kit and RNA quality was assessed using the Agilent Bioanalyzer 6000 Pico Kit [RNA Integrity Number (RIN) = 7.68 ± 0.26 (mean ± s. d.)].

## RNA sequencing and analysis

Three sequencing libraries were prepared from each RNA sample using 5 ng total RNA in the SMART-Seq v4 Ultra Low Input RNA Kit (Takara), according to the manufacturer's protocol. The cDNA was fragmented and tagged with sequencing adapters using Nextera XT kit (Illumina). The transcripts were quantified with the KAPA qPCR complete kit (BioRad) for Illumina platforms. RNA libraries were sequenced on llumina HiSeq 4000 platform. Library preparation and sequencing were performed by the Northwest Clinical Genomics Laboratory at the University of Washington. We aligned reads to the mm10 transcriptome using the salmon aligner v0.11.3 (*Patro et al., 2017*), and then imported into R and summarized at the gene level using the Bioconductor tximport package, v1.8.0 (*Soneson et al., 2015*). We then filtered out any genes with consistently low counts (retaining those genes with >10 counts in at least three samples). After filtering, 19,339/35,728 (54%) genes remained. Differential gene expression was performed using the Bioconductor edgeR package v3.22.3 (*Robinson et al., 2010*), implemented in the R programming language. We fit the model and then made comparisons using quasi-likelihood F-tests. We incorporated a log fold change >±0.263 (representing a 20% change in expression) as part of the comparison, using the glmTreat function in edgeR (*McCarthy and Smyth, 2009*), and selected genes with a false discovery rate (FDR) < 0.05. We then performed a set of self-contained gene set tests based on the PI3K pathway (GO:0043491), YAP pathway (GO:0035329), and the genes that are differentially expressed between the developing gyrus and sulcus in ferret (*de Juan Romero et al., 2015*). To generate the gene set based on ferret brain, we downloaded and processed data from the Gene Expression Omnibus (GSE60687), using the Bioconductor GEOquery v2.48.0 and limma packages, v3.36.2 (*Davis and Meltzer, 2007*; *Ritchie et al., 2015*), identifying 362 genes with FDR < 0.1. We used NCBI BLAST +to align probe sequences to the RefSeq database, then matched gene symbols to our mouse RNA-seq dataset, resulting in 168 matching genes. We generated barcode plots to help visualize results from the gene set tests using functions in the limma package. Bar graph was made in Microsoft Excel using log transformed counts, after normalizing for library size (log counts/million counts). Venn diagram was made using the online application http://bioinformatics.psb.ugent.be/webtools/Venn/.

## Quantitative analysis

Number of mice is consistent with previous experiments completed and published by us and other investigators and based on power analyses. For histology and length measurement experiments, we performed power analyses in R using the pwr package. We used preliminary data to estimate sample variance and calculated an effect size of 1.8 between groups. We then used this effect size of 1.8 to estimate that a sample size of 5 animals per group will be required to have 80% power to detect significant differences (p=0.05) between groups. Based on this, we considered group size of 5–8 mice/genotype for each experiment to be sufficient, unless otherwise specified; extra mice/cells were considered for possible technical issues. For quantitative analysis of embryos, data was collected from comparable sections of each genotype/condition (from two or more independent litters) at each developmental stage. All measurements were made using ImageJ and ImageJ2 software programs (NIH, Bethesda, Maryland, USA). Medial tissue length was measured in the lateral ventricular lining from the dorso-medial cortical notch to the fimbrial tip (*Roy et al., 2015*); the data was normalized to the control value (n = 8/genotype). P3 critical period CA1 length was measured along the mid-

thickness of the PL of each mutant type; data was normalized to the P3 CA1 length (induced E0.5>P0). Cell counts from E14.5, E16.5 and P0/P3 brains were obtained from the middle part of the CA1 dorso-ventral extent. To avoid counting discrepancies related to decreased cell density in E16.5 *Pik3ca* mutant (*Roy et al., 2015*), all E16.5 cell counts were done by dividing each cell type by the total number of DAPI$^+$ cells present within the fixed area of quantitation. Postnatal nYap$^+$ cell distribution quantitation was done by binning the CA1 region into mono-layer apical edge, ventricular-subventricular zone (vz/svz) and the remaining area that includes the white matter (*stratum oriens*) and the pyramidal layer. Confocal stacks of immuno-stained sections of each developmental stage were generated by scanning at intervals of 0.99 μm using filters of appropriate wavelengths at 20X, 40X, 63X and 100X magnifications. Representative images for cell adhesion molecules corresponded to one 0.99μm-thick confocal plane. Measurements for labeling index (n = 7/genotype), quit fraction (n = 12/genotype), birth-dating studies (n = 5/genotype), TUNEL cell counts (n = 7 hemi-sections/ genotype/condition), P0 Tbr2$^+$ basal progenitor cell counts (n = 7/genotype/condition) were calculated using ImageJ. Statistical significance was assessed using 2-tailed unpaired t-tests with Welch's correction (for medial length measurements, labeling index, quit fraction and progenitor cell counts, birth-dating experiments) and ANOVA followed by Tukey post-test (one-way ANOVA for critical period CA1 length, TUNEL counts and total P3 nYap$^+$ cell counts; two-way ANOVA for P0 total nYap$^+$ and Tbr2$^+$ cell counts, and nYap$^+$ cell distribution post-verteporfin treatment). These analyses were performed in GraphPad Prism v7.0 (GraphPad Software Inc, San Diego, USA) and in Microsoft Excel. Differences were considered significant at p<0.05. Data are represented as mean ± SEM for *Figures 2*, *5* and *7*, *Figure 1—figure supplement 3*, *Figure 2—figure supplement 1*, *Figure 3—figure supplement 1* and *Figure 7—figure supplement 2*. ARRIVE guidelines have been followed for reporting work involving animal research.

## Acknowledgements

We thank Jean J Zhao for gifts of mouse lines (*GFAP-cre* and *Rosa26-rtTA; Pik3ca$^{H1047R}$*); Paul Wakenight and William B Dobyns for discussions. The mouse anti-Pax6 antibody developed by Kawakami A, was obtained from the Developmental Studies Hybridoma Bank (DSHB), created by the NICHD of the NIH and maintained at The University of Iowa, Department of Biology, Iowa City, IA 52242.

## Additional information

### Funding

| Funder | Grant reference number | Author |
|---|---|---|
| National Institutes of Health | 1R01NS099027 | Kathleen J Millen |
| Seattle Children's Hydrocephalus Research Guild | Seed fund | Kathleen J Millen |
| Eunice Kennedy Shriver National Institute of Child Health and Human Development | U54HD083091 | Theo K Bammler |

The funders had no role in study design, data collection and interpretation, or the decision to submit the work for publication.

### Author contributions

Achira Roy, Conceptualization, Formal analysis, Validation, Investigation, Visualization, Methodology, Writing—original draft, Project administration, Writing—review and editing; Rory M Murphy, Formal analysis, Visualization; Mei Deng, Performed laser-capture microdissection; James W MacDonald, Data curation, Formal analysis, Contributed to RNA-seq analysis; Theo K Bammler, Formal analysis, Funding acquisition, Contributed to RNA-seq analysis; Kimberly A Aldinger, Data curation, Validation, Contributed to RNA-seq data analysis and interpretation; Ian A Glass, Resources; Kathleen J Millen, Conceptualization, Formal analysis, Supervision, Funding acquisition, Project administration, Writing—review and editing

Author ORCIDs
Achira Roy https://orcid.org/0000-0002-6274-0667
Kimberly A Aldinger https://orcid.org/0000-0002-5406-8911
Kathleen J Millen https://orcid.org/0000-0001-9978-675X

Ethics
Animal experimentation: Animal experimentation: All animal experimentation was conducted in accordance with the guidelines laid down by the Institutional Animal Care and Use Committees (IACUC) of Seattle Children's Research Institute, Seattle, WA, USA (protocols 14208 (008) and 14395 (006)).

Decision letter and Author response
Decision letter https://doi.org/10.7554/eLife.45961.034
Author response https://doi.org/10.7554/eLife.45961.035

## Additional files

### Supplementary files
• Transparent reporting form
DOI: https://doi.org/10.7554/eLife.45961.026
• Reporting standard 1. The ARRIVE guidelines checklist.
DOI: https://doi.org/10.7554/eLife.45961.027

### Data availability
RNA-seq data have been deposited in the NCBI Gene Expression Omnibus under the accession code GSE127896. Related analysed data are provided in Figure 6—source data 1 and Figure 6—source data 2 for Figure 6 and Figure 6—figure supplements 2 and 3.

The following dataset was generated:

| Author(s) | Year | Dataset title | Dataset URL | Database and Identifier |
| --- | --- | --- | --- | --- |
| Achira Roy, Rory M Murphy, Mei Deng, James W MacDonald, Theo K Bammler, Kimberly A Aldinger, Ian A Glass, Kathleen J Millen | 2019 | PI3K-Yap activity drives cortical gyrification and hydrocephalus in mice | https://www.ncbi.nlm.nih.gov/geo/query/acc.cgi?acc=GSE127896 | NCBI Gene Expression Omnibus, GSE127896 |

The following previously published dataset was used:

| Author(s) | Year | Dataset title | Dataset URL | Database and Identifier |
| --- | --- | --- | --- | --- |
| Romero CD, Bruder C, Martínez-Martínez MA, Tomasello U, Sanz-Anquela JM, Borrell V | 2015 | Sharp changes in gene expression levels along germinal layers distinguish the development of gyrencephaly | https://www.ncbi.nlm.nih.gov/geo/query/acc.cgi?acc=GSE60687 | NCBI Gene Expression Omnibus, GSE60687 |

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
