## [Decision Letter]

Thank you for submitting your article "PI3K-Yap activity drives cortical gyrification and hydrocephalus in mice" for consideration by *eLife*. Your article has been reviewed by three peer reviewers, and the evaluation has been overseen by a Reviewing Editor and Sean Morrison as the Senior Editor. The reviewers have opted to remain anonymous.

The reviewers have discussed the reviews with one another and the Reviewing Editor has drafted this decision to help you prepare a revised submission.

Summary:

This paper follows up on an earlier study published in *eLife* in 2015 that showed that mice with constitutive cortical expression of PI3K present cortical dysplasia and other pathological features seen in patients with intractable pediatric epilepsy. In this new paper, the authors show that the same mice present hydrocephalus and aberrant folding of the hippocampus which can be triggered by a brief activation of the pathway during embryonic development and can be attenuated by maternal delivery of the Yap inhibitor, verteporfin. They link the generation of the folds to Yap nuclear translocation and changes in apical junctions and they present the findings as evidence that regulation of apical cell adhesion of cortical progenitors underlie the initiation of cortical gyrification.

All of the reviewers find this study interesting and convincing and the experimental data of excellent quality. However there is a consensus that while this paper is framed around the idea that PIK3CA mice constitute a model of cortical gyrification, the study in fact focuses on the defects in the hippocampus, with limited analysis and illustration of the neocortical phenotype. The data presented are not sufficient to claim that the neocortical defects reflect a process of gyrification. Moreover, it is unclear whether the much more dramatic folds of the hippocampus also represent gyrification. To address these and other concerns, the reviewers request that the following essential revisions are made to the manuscript:

1) The neocortex of PIK3CA mice should be further analysed and illustrated to provide stronger evidence of a gyrification process, or if the data are not conclusive, substantial changes should be made to the title, Abstract and text. Information on the penetrance of the phenotype and higher magnification pictures should be provided.

2) The hippocampus of PIK3CA mice should also be further analysed and illustrated to provide more convincing evidence of a gyrification process. For example, are there differences in thickness of the putative gyri and sulci, as in a gyrated cortex?

3) Some explanation should be provided of why the hippocampus folds much more extensively than the cortex, and why there is a posterior to anterior gradient in folding of the hippocampus. Does this reflect a gradient in PiK3CA expression levels (which should be illustrated), differences in nuclear YAP levels or in apical junction organisation, etc.?

4) There is a discrepancy between the decrease in pHH3 expression observed in E14.5 brains and the comment (subsection “Focal increases in progenitors initiated bona fide cortical gyrification in *Pik3ca^H1047R^* mutant”, second paragraph) that there is higher proliferation and expansion of apical progenitors. This should be explained or corrected.

5) Cell death could be analysed in the hippocampus of PIK3CA mice without and with verteporfin treatment to determine whether this contributes to the phenotypes.

6) The nuclear localisation of YAP should be better illustrated in Figure 5, particularly at E14.5 and E16.5, including with higher magnification images. The data in Figures 5 and 7 should be quantified and analysed statistically. The number of brains analysed after verteporfin treatment and the number of brains showing an attenuation of the phenotype should be provided.

7) Hydrocephalus should be better documented. How penetrant is this phenotype? Can it be related to a structural or cellular defect in the transgenic brains (e.g. stenosis, abnormal ependymal layer)?

---

## [Author Response]

Summary:[…] All of the reviewers find this study interesting and convincing and the experimental data of excellent quality. However there is a consensus that while this paper is framed around the idea that PIK3CA mice constitute a model of cortical gyrification, the study in fact focuses on the defects in the hippocampus, with limited analysis and illustration of the neocortical phenotype. The data presented are not sufficient to claim that the neocortical defects reflect a process of gyrification. Moreover, it is unclear whether the much more dramatic folds of the hippocampus also represent gyrification.

As noted below, we have added additional data and arguments demonstrating that cortical gyrification in the *Pik3ca* mutant mice is *cre*-dependent, and that both the neocortical and hippocampal tissue folding have all the hallmarks of bona fide cortical gyrification.

To address these and other concerns, the reviewers request that the following essential revisions are made to the manuscript:1) The neocortex of PIK3CA mice should be further analysed and illustrated to provide stronger evidence of a gyrification process, or if the data are not conclusive, substantial changes should be made to the title, Abstract and text. Information on the penetrance of the phenotype and higher magnification pictures should be provided.

Similar to our mutant phenotype (this study; Roy et al., 2015) a study by another group using a separately generated *Pik3ca^H1047R^* mouse line under *Emx1-cre* driver, demonstrated almost identical gyrification phenotypes, in both hippocampus and neocortex (D'Gama et al., 2017). Please refer to Figure 4 of D'Gama et al. (2017), excerpt from the figure legend says: “…(A–D) Compared with the wild-type P7 cortex (A and C), the *Emx1-Cre;Pik3ca^H1047R^*/wt P7 cortex (B and D) is larger, with marked gyrification in the cingulate cortex, neocortex, and piriform cortex (arrows in B and D)…”. This proves that the neocortical and hippocampal gyrification phenotypes are reproducible and generalizable, even across different *Pik3ca^H1047R^* transgenic mouse lines under different *cre* drivers. There was minimal analysis of the gyrification phenotype in this other paper.

In our model, there is 100% penetrance in the gyrification phenotype both in hippocampus and neocortex. The extent and positioning of folds in the neocortex are variable however. We have now provided a better characterization of the *Pik3ca* mutant neocortex, including the information on phenotypic penetrance and magnified images of the folds. See Figure 1—figure supplement 2A-J and Results and Discussion sections. Both neocortical and hippocampal folds meet all the criteria for bona fide cortical gyrification, defined in (Borrell, 2018), and this is now reiterated in the text. Point to be noted is that we have previously published that there is abnormal neuronal migration in the *hGFAP-cre;H1047R* mouse neocortex (Roy et al., 2015); hence identification of certain neocortical layers is not readily possible.

2) The hippocampus of PIK3CA mice should also be further analysed and illustrated to provide more convincing evidence of a gyrification process. For example, are there differences in thickness of the putative gyri and sulci, as in a gyrated cortex?

True stereotypic cortical folding is characterized by the folding of the pial surface, the underlying neuronal layers and pial/basal side of the white matter, whereas the ventricular surface remains smooth (Borrell, 2018). We have now provided new data and illustration of the folding and of the differential thickness across *Pik3ca* mutant hippocampal gyri and sulci in Figure 1—figure supplement 2K-T.

3) Some explanation should be provided of why the hippocampus folds much more extensively than the cortex, and why there is a posterior to anterior gradient in folding of the hippocampus. Does this reflect a gradient in PiK3CA expression levels (which should be illustrated), differences in nuclear YAP levels or in apical junction organisation, etc.?

We have explained this in the Results section. The *hGFAP-cre* driver used in our studies (this study; Roy et al., 2015) is active only from ~embryonic day (E)13, the time when neurogenesis in neocortex is already at its peak. Moreover, the *hGFAP-cre* line has a strong high-medial-low-lateral expression gradient. Even within the lateral neocortex, especially around E14.5 (Figure 1—figure supplement 1C, D), an apical-low-basal-high gradient is observed. Thus, we think that the time of initiation and spatio-temporal gradients of *cre* expression pattern explains the differential folding between neocortex and hippocampus.

The size of hippocampus normally is smaller in anterior coronal sections, compared to posterior sections; hence in the mutant we see more hippocampal folds in the posterior sections. We find no evidence for any gradients in YAP or apical junction organization.

4) There is a discrepancy between the decrease in pHH3 expression observed in E14.5 brains and the comment (subsection “Focal increases in progenitors initiated bona fide cortical gyrification in Pik3ca^H1047R^ mutant”, second paragraph) that there is higher proliferation and expansion of apical progenitors. This should be explained or corrected.

We have now corrected the E14.5 BrdU/pHH3 images since the original mutant image was not representative of the entire mutant cohort; we thank the reviewers for pointing this out. Our comment on proliferation is based on the short-pulse BrdU experiments and calculation of labeling index for each embryonic age.

5) Cell death could be analysed in the hippocampus of PIK3CA mice without and with verteporfin treatment to determine whether this contributes to the phenotypes.

We have previously published that TUNEL^+^ cell number was significantly lower in E16.5 *Pik3ca^H1047R^* mutant neocortex than in control (p < 0.01), indicating reduced apoptosis (Figure 3—figure supplement 1C in Roy et al., 2015) However the overall TUNEL^+^ cell numbers for both control and mutant were small, indicating that cortical expansion in the mutant was not primarily driven by reduced apoptosis.

We now specifically conducted the TUNEL experiment in E16.5 control and *Pik3ca* mutant (+/- verteporfin) and see no significant differences in the TUNEL^+^ cell number in the hippocampal CA1 region across groups (Figure 7—figure supplement 2H). We conclude that cell death does not contribute to the phenotypes.

6) The nuclear localisation of YAP should be better illustrated in Figure 5, particularly at E14.5 and E16.5, including with higher magnification images. The data in Figures 5 and 7 should be quantified and analysed statistically. The number of brains analysed after verteporfin treatment and the number of brains showing an attenuation of the phenotype should be provided.

We have now provided images of higher magnification at E14.5 and E16.5 in Figure 5. We have also added quantification of the total number and distribution of nYap^+^ cells in the CA1 region in Figures 5 and 7. Additional details of verteporfin experiments are now further elaborated in the Materials and methods section. We used 7 mice/genotype/treatment across 5 P0 litters; 100% of the mutant brains demonstrated attenuation of the gyrification and hydrocephalus phenotypes.

7) Hydrocephalus should be better documented. How penetrant is this phenotype? Can it be related to a structural or cellular defect in the transgenic brains (e.g. stenosis, abnormal ependymal layer)?

We have now clarified these points in the text. The hydrocephalus phenotype in *hGFAP-cre;Pik3ca^H1047R^*mutant mice is 100% penetrant. No stenosis has been observed in any of the mutants. Our study clearly demonstrates that defects in early ependymal development correlate with the mutant ventriculomegaly/ hydrocephalus, which in turn, was rescued by normalizing the apical ependymal/ventricular edge with verteporfin treatment.